# Revisiting Neighbourhoods in Mean Field Reinforcement Learning

**Sriram Ganapathi Subramanian***                    *sriramsubramanian@cunet.carleton.ca*
*Carleton University*
*Vector Institute for Artificial Intelligence*

**Matthew E. Taylor**                                 *matthew.e.taylor@ualberta.ca*
*University of Alberta*
*Alberta Machine Intelligence Institute*

**Kate Larson**                                       *kate.larson@uwaterloo.ca*
*University of Waterloo*

**Mark Crowley**                                      *mark.crowley@uwaterloo.ca*
*University of Waterloo*

**Pascal Poupart**                                    *pascal.poupart@uwaterloo.ca*
*University of Waterloo*
*Vector Institute for Artificial Intelligence*

**Reviewed on OpenReview:** *https://openreview.net/forum?id=PQ5R7K0WDc&noteId=yoAoNIUbf9*

## Abstract

Many multi-agent reinforcement learning (MARL) algorithms do not scale well as the number of agents increases due to an exponential time and space complexity dependency on the number of agents in the environment. Mean field theory has been used to address this problem by approximating the effect of neighbourhoods of agents by a single representative agent. While this approximation allows MARL algorithms to scale to environments with many agents, approaches typically assumed that agents 1) inside a neighbourhood are homogeneous, and 2) outside a neighbourhood have no influence (and can therefore be ignored). This paper relaxes these assumptions and proposes a novel framework, mean field attention (MFA), which uses an attention mechanism for local responses and the mean field approximation for global responses. We implement MFA with two new algorithms leveraging Q-learning and actor-critic. These novel MFA algorithms consistently outperform other MARL algorithms, including prior mean field-based algorithms, across multiple metrics and benchmarks.

## 1 Introduction

Multi-agent reinforcement learning (MARL) has seen many breakthroughs in the past decade (Hernandez-Leal et al., 2019). However, almost all MARL applications involve environments with a few agents (tens or fewer). It has been difficult to replicate these successes in environments with many agents, as algorithms typically have an exponential dependence on the number of agents in terms of space and time complexity (Busoniu et al., 2008). One solution is to use mean field theory (Stanley, 1971) to scale MARL. Mean field theory

---

*Corresponding Author

approximates the effect of all other agents in the environment using a single averaged effect (called the *mean field*), which reduces a many-agent setting into a two-agent setting. This mean field approximation has been successfully used by prior works to make MARL tractable in many agent environments (Lasry & Lions, 2007; Yang et al., 2018).

Yang et al. (2018) introduced a mean field reinforcement learning (MFRL) framework that integrates the mean field approximation into stochastic games by using the empirical action distribution of all agents in the environment as the mean field. This framework contains three significant limitations that prevent its wide application. First, MFRL assumes the presence of a fixed neighbourhood around each individual agent. For a representative agent (i.e., a *central agent*), all agents within this neighbourhood are required to be homogeneous and have the same level of importance in the value function of the central agent (they are approximated using the mean field). Second, the identity of agents within the neighbourhood are determined a priori and cannot be changed during training. Third, all agents outside the neighbourhood must have no impact on the central agent (they can be ignored). However, in practice, different agents within the neighbourhood can have different impact on the central agent, agents can move in and out of the neighbourhood, and agents outside the neighbourhood can have (some) impact on the central agent. Most importantly, the notion of neighbourhoods has a fundamental disconnect with the mean field approximation. On the one hand, the mean field approximation is used to make MARL tractable in environments with very large numbers of agents, and on the other hand, neighbourhoods restrict the number of agents in the environment (i.e., agents outside the neighbourhood are ignored). Hence, when neighbourhoods are used, it is often practical to simply use multi-agent algorithms directly without a need for the mean field approximation (since the number of agents that matter is already highly reduced).

Furthermore, the mean field provides an approximation that, if applied across all agents, provides a global signal regarding the aggregated behaviour of all agents. While this global signal is helpful for long-term strategy and planning, agents must also prioritize nearby agents for effective short-term decision-making. For example, in autonomous driving, an agent can choose its route to the destination using the mean field which will naturally capture areas of high traffic, however, to avoid collisions, it must give greater importance to nearby vehicles.

We relax the notion of neighbourhoods and introduce a novel MFRL framework, *mean field attention* (MFA), that uses an attention mechanism (Vaswani et al., 2017) to dynamically attend to different agents within a fixed importance set (a finite set of agents close to the central agent), and then leverages the mean field to approximate the effect of agents outside the importance set. The key insight is that attention can effectively capture context, allowing it to zero in on the most relevant agents (Brauwers & Frasincar, 2022). MFA addresses the three MFRL limitations. First, MFA does not assume that agents within an importance set will have the same impact, instead it uses the attention mechanism to weight them differently. Second, while the size of the importance set used by MFA needs to be determined a priori, the identity of agents can change during training/execution. Third, MFA ensures all agents are considered, applying the mean field approximation to those that fall outside the designated importance set. We present two practical MFA algorithms, show that it is possible to extend MFA to heterogeneous environments having distinct types of homogeneous agents (Subramanian et al., 2020), and provide techniques that accelerate MFA training using external knowledge sources. Through detailed experiments, we demonstrate that our methods scale well and provide better performances than several state-of-the-art baselines. For the first time, we extend MFRL to real-world environments that do not have well-defined neighbourhoods of homogeneous agents.

We list the major contributions made by this paper to the literature in scaling MARL to many agent environments:

1. The neighbourhood assumptions in prior MFRL methods have a fundamental disconnect with the mean field approximation, as explained above. This paper contributes a new framework, MFA, that provides a principled relaxation of this assumption. Furthermore, this paper provides two practical algorithms using the MFA framework, one using a value-based (Watkins & Dayan, 1992) approach (Mean Field Attention Q-Learning, abbreviated as MFA-QL) and the other using a policy gradient-based (Sutton et al., 1999) approach (Mean Field Attention Actor-Critic, abbreviated as MFA-AC).

2. For the first time, this paper conducts a large-scale experimental study of MARL algorithms in various cooperative, competitive, and mixed-motive many-agent environmental scenarios containing hundreds of agents (for instance, some environments using the MAgent (Zheng et al., 2017) testbed contain a total of 288 agents, with 144 agents per team; see Section 5).

3. This paper provides an extensive benchmarking effort establishing the performances of independent algorithms, centralized training methods, mean field methods, and attention-based approaches in many-agent environments (see Section 5 and Appendix B). Several novel insights from these extensive experiments are provided, which form an important contribution to the MARL literature.

4. This paper demonstrates that it is possible to extend the MFA framework to environments containing heterogeneous agents and also accelerate training using pre-existing sources of knowledge (for which separate practical algorithms are provided). This is the first mean field algorithm to accelerate training using external knowledge sources, which leads to wide improvements in sample efficiency, as shown in the experiments (see Section 5).

5. True to the stated objective of deploying mean field methods in practical applications, for the first time, this paper applies MFA algorithms (and other baseline mean field methods) to a complex real-world environment (autonomous driving), where agents learn to perform extremely complicated manoeuvres (such as lane changing and navigating intersections) to achieve goals.

In this work, we also provide theoretical results showing that MFA retains the important theoretical guarentees of prior MFRL methods (see Appendix G). Nonetheless, the primary contributions of this paper are methodological and empirical as listed in the *major contributions* above.

## 2 Background

**Stochastic Games**: We use the stochastic game setting represented as $\langle S, A^1, \ldots, A^N, r^1, \ldots, r^N, p, \gamma \rangle$, where $N$ is the number of agents, $S$ is the (global) state space, $A^j$ is the action space of an agent $j$, $r^j : S \times A^1 \times \cdots \times A^N \to \Re$ is the reward function of the agent $j$, and $p : S \times A^1 \times \cdots A^N \to \Omega(S)$ is the transition probability. $\gamma \in [0, 1)$ is the discount factor. In a stochastic game, each agent observes the global state $s$ at the current time $t$ and takes a local action $a^j$ (Shapley, 1953). The joint action of all agents, $\boldsymbol{a} = \{a^1, \ldots, a^N\}$, determines the immediate reward $r^j$ for the agent $j$ and the next state $s'$. The objective of each agent is to learn a policy that gives the best responses to other agents and the environment. The policy of an agent $j$ is denoted by $\pi^j : S \to \Omega(A^j)$. Let $\boldsymbol{\pi} = \{\pi^1, \ldots, \pi^N\}$ represent a time-independent joint policy of all agents. At a state $s$, the value function of the agent $j$ under the joint policy $\boldsymbol{\pi}$ is the expected reward of the agent $j$, when all agents follow $\boldsymbol{\pi}$ from $s$, expressed as $v^j(s; \boldsymbol{\pi}) = \sum_{t=0}^{\infty} \gamma^t \mathbb{E}_{\boldsymbol{\pi}, p}[r_t^j | s_0 = s, \boldsymbol{\pi}]$. An alternate representation to the value function is the $Q$-function, $Q_{\boldsymbol{\pi}}^j : S \times A^1 \times \cdots \times A^N \to \Re$, expressed as $Q_{\boldsymbol{\pi}}^j(s, \boldsymbol{a}) = r^j(s, \boldsymbol{a}) + \gamma \mathbb{E}_{s' \sim p}[v^j(s'; \boldsymbol{\pi})]$ for the agent $j$ under $\boldsymbol{\pi}$. Since the $Q$-function depends on the joint action, most MARL algorithms have an exponential dependence on the number of agents (Busoniu et al., 2008).

**Nash $Q$-value**: In the stochastic game setting, the *Nash equilibrium* (NE) is typically considered as the solution concept (Hu & Wellman, 2003), represented by the joint policy $\boldsymbol{\pi}_* = [\pi_*^1, \ldots, \pi_*^N]$, such that for all $s \in S$, all $\pi^j$, and all $j$, $v^j(s; \pi_*^j, \boldsymbol{\pi}_*^{-j}) \geq v^j(s; \pi^j, \boldsymbol{\pi}_*^{-j})$ is satisfied. Here the notation $\boldsymbol{\pi}^{-j} \triangleq [\pi_*^1, \ldots, \pi_*^{j-1}, \pi_*^{j+1}, \ldots, \pi_*^N]$ represents the joint policy of all agents except that of agent $j$. In other words, in a NE, each agent plays its best response to the other agents, and any unilateral deviation from this response is guaranteed to be worse off. The $Q$-value of $j$ when all agents play the NE policy (i.e., the Nash $Q$-value) can be expressed as $Q_*^j(s, \boldsymbol{a}) = r^j(s, \boldsymbol{a}) + \gamma \sum_{s' \in \mathcal{S}} p(s'|s, \boldsymbol{a}) v^j(s'; \pi_*^1, \ldots, \pi_*^n)$.

**MFRL**: With assumptions of locality (neighbourhoods) and homogeneity, Yang et al. (2018) proved that the multi-agent $Q$-function can be replaced by a mean field $Q$-function (MFQ), $Q^j(s_t, \boldsymbol{a}_t) = Q^j(s_t, a_t^j, \bar{a}_t^j)$. The MFQ does not depend on the joint action space and has a constant dependence on the number of agents. It is recurrently updated using Eqs. 1 – 4. Here, $r_t^j$ represents the reward for the agent $j$, $s_t$ and $s_{t+1}$ represent the state at time steps $t$ and $t + 1$, respectively, $\alpha$ is the learning rate, $v^j(s_t)$ represents the value function of the agent $j$ at state $s_t$, and $\beta$ is the Boltzmann parameter. Notably, $a_t^j$ denotes the (discrete) action of

the agent $j$ represented as a one-hot encoding whose components are one of the actions in the action space. The notation $\overline{a}_t^j$ represents the mean action of all other agents apart from the agent $j$ (i.e., $\overline{a}_t^j$ is the mean field at time $t$) and $\pi^j$ denotes the Boltzmann policy of the agent $j$. In Eq. 2, there is no expectation over $\overline{a}^j$, because Yang et al. (2018) guaranteed that the MFQ updates will be greedy in the limit ($t \to \infty$) with infinite exploration (GLIE).

$$Q^j(s_t, a_t^j, \overline{a}_t^j) = (1 - \alpha)Q^j(s_t, a_t^j, \overline{a}_t^j) + \alpha[r_t^j + \gamma v^j(s_{t+1})] \tag{1}$$

$$\text{where } v^j(s_{t+1}) = \sum_{a_{t+1}^j} \pi^j(a_{t+1}^j | s_{t+1}, \overline{a}_t^j) Q^j(s_{t+1}, a_{t+1}^j, \overline{a}_t^j) \tag{2}$$

$$\overline{a}_t^j = \frac{1}{N} \sum_{k \neq j} a_t^k, a_t^k \sim \pi^k(\cdot | s_t, \overline{a}_{t-1}^k) \tag{3}$$

$$\text{and } \pi^j(a_t^j | s_t, \overline{a}_{t-1}^j) = \frac{\exp(-\beta Q^j(s_t, a_t^j, \overline{a}_{t-1}^j))}{\sum_{a_t^{j'} \in A^j} \exp(-\beta Q^j(s_t, a_t^{j'}, \overline{a}_{t-1}^j))}. \tag{4}$$

Finally, Yang et al. (2018) provide two algorithms, mean field $Q$-learning (MFQ) and mean field actor-critic (MFAC) that use function approximation for the $Q$-functions and a replay buffer (Mnih et al., 2015).

The MFRL framework of Yang et al. (2018) relies on locality and homogeneity assumptions (comprising the "neighbourhood" assumptions). The locality assumption states that each agent $j$ interacts only with agents within a fixed, pre-defined neighbourhood, and agents outside this neighbourhood are assumed to have no influence on agent $j$ and are ignored. The homogeneity assumption states that all agents within the neighbourhood are assumed to be homogeneous (i.e., they have the same state space, action space, and reward functions) and identical in their impact on the central agent, such that their collective effect can be summarized by a single mean action. Formally, including these assumptions in Yang et al. (2018)'s framework, for a central agent $j$ with neighbourhood $\mathcal{N}^j$, the mean field is computed as according to Eq. 15 (where $k \in \mathcal{N}^j$) and agents outside $\mathcal{N}^j$ are excluded from this computation entirely.

## 3    Related Work

**Scaling MARL**: The simplest technique for scaling MARL is independent learning (Tan, 1993) which assumes all other agents to be part of the environmental state and directly uses single-agent approaches in multi-agent environments. This method has not been very successful since the underlying Markovian assumptions of these algorithms are violated (Lee et al., 2022). There are three different mean field paradigms in the literature for scaling MARL: Mean Field Games (Lasry & Lions, 2007), Mean Field Control (Bensoussan et al., 2013), and MFRL (Yang et al., 2018), each with its own unique problem formulations, assumptions, approximations, solution concepts, associated learning algorithms, and distinct applications. Our work is based on the MFRL paradigm. While effective, MFRL relies on two important assumptions regarding the homogeneity of agents and the presence of pre-defined neighbourhoods. The homogeneity assumption has been relaxed by Subramanian et al. (2020), which extended MFRL to environments with heterogeneous agents, where agents can be grouped into a finite set of types (with homogeneity maintained within a type, but not across types). However, the assumptions of locality (neighbourhoods) have been retained by previous MFRL methods. We relax this in our work.

**Attention Mechanism**: The attention mechanism (Vaswani et al., 2017) mimics the differentiable key-value memory model typically used in database retrieval (Graves et al., 2014; Oh et al., 2016). In this context, a powerful attention based MARL method is the multi-actor-attention-critic (MAAC) that uses a central critic relying on the attention mechanism to extract relevant information about all other agents at each time step (Iqbal & Sha, 2019). MAAC still has a quadratic dependence on the number of agents, which contrasts with mean field approaches that provide a constant dependence. Also, since the MAAC attention module includes inputs from all agents for decision-making, it is computationally too demanding in many-agent environments. Further, the attention mechanism has been explored in MFRL previously. Wang et al. (2022)

provide the weighted MFRL (WMFRL) method that derives a weighted mean field approximation and uses an attention mechanism within the neighbourhood to determine the weights of this mean field. Independently, Wang et al. (2024) also provide a very similar technique, called adaptive MFQ (AMFQ) that uses an attention mechanism to determine a weighted mean field. In our experiments, we found that WMFRL performs much better than AMFQ, since it uses a double $Q$-learning update (Hasselt, 2010). Hao et al. (2023) propose the Graph Attention Mean Field (GAT-MF) technique, that uses a graph attention mechanism (Veličković et al., 2017) to determine the weights of the weighted mean field in a neighbourhood of adjacent agents. GAT-MF is restricted to environments with agents having fixed relative positions, where a static graph can model the adjacency. Notably, all of these attention MFRL methods retain the neighbourbood assumption from Yang et al. (2018) and ignore agents outside the neighbourhood. Seperately, Yu (2023) provide the hierarchical mean field (HMF) technique that divides all agents into a finite set of groups. Two levels of learning is performed: bottom level for intra-group and top level for inter-group interactions. The bottom level uses an attention mechanism and a mean field approximation.

## 4   Mean Field Attention

MFA brings three novelties to MFRL: 1) the introduction of *importance sets*, 2) accounting for all agents in the environment through mean field approximation outside the importance set (global responses), and 3) incorporating the attention mechanism for determining different levels of importance for other agents inside the importance set (local responses). As in MFRL (Yang et al., 2018), we assume that all agents have a discrete action space. Initially we assume that the local observation space and the action space are the same for all agents (this is a partial homogeneity assumption), and relax this at the end of this section (also see Appendix J for more details).

For a representative agent $j$ (i.e., the central agent), the importance set (denoted by $\mathcal{I}^j$), contains a set of $K^j$ ($K^j = |\mathcal{I}^j|$) agents that have the most influence on the decision making of the agent $j$. Before training, MFA requires the specification of $K^j$ for each $j$. The importance set consists of the $K^j$ agents closest to the central agent $j$, determined by a notion of distance. In this paper, we use the Euclidean distance between agents in the environment's state space as the distance metric across all experimental domains considered in Section 5. We note that MFA is a general framework that can accommodate any domain-appropriate notion of distance or proximity, for example, strategies of for-profit industries are highly influenced by a set of close competitors who can form the importance set (i.e., near-in-the-sense of competition). We assume that such a distance metric is available for the domain based on which the importance set is defined (we discuss settings where this may not hold in Appendix A). We use the variable $N$ to denote the total number of agents (where $N$ can be large in accordance with a mean field formulation). Every agent considers a set of $K^j$ other agents nearby as part of its importance set (indexed by $i \in [1, \ldots, K^j]$), where $K^j$ is expected to be smaller than $N$. $K^j$ is a hyperparameter, which can be determined per agent using domain knowledge.

We note that both the distance metric and the choice of $K^j$ are domain-dependent hyperparameters: while physical Euclidean distance serves as a natural metric in the environments considered in our paper (see Section 5), domains lacking a natural notion of proximity may require learned distance metrics, and the appropriate value of $K^j$ should be determined using domain knowledge (see Appendix A and Appendix B.1).

While importance sets may seem similar to neighbourhoods in MFRL, there are three key differences. First, though the importance set size is determined a priori, the identity of agents in the importance set of the central agent is not fixed. Other agents can move in and out of the importance set. Second, agents within the importance set need not have the same impact on the central agent. Third, the mean field approximation is applied to agents outside the importance set and no agents are ignored.

A natural question is whether the mean field approximation outside the importance set is necessary at all. One might hypothesize that distant agents already have negligible influence and can simply be ignored. Recent work by Qu et al. (2022) shows that under a "local interaction structure," defined by a fixed dependence graph $G$ where each agent's transition distribution depends only on its local graph neighbourhood, the Q-function satisfies an exponential decay property with respect to graph distance. However, this result is explicitly restricted to networked system applications such as wireless communication and social networks that possess this fixed graph structure. In the dynamic environments targeted by MFA, such as autonomous

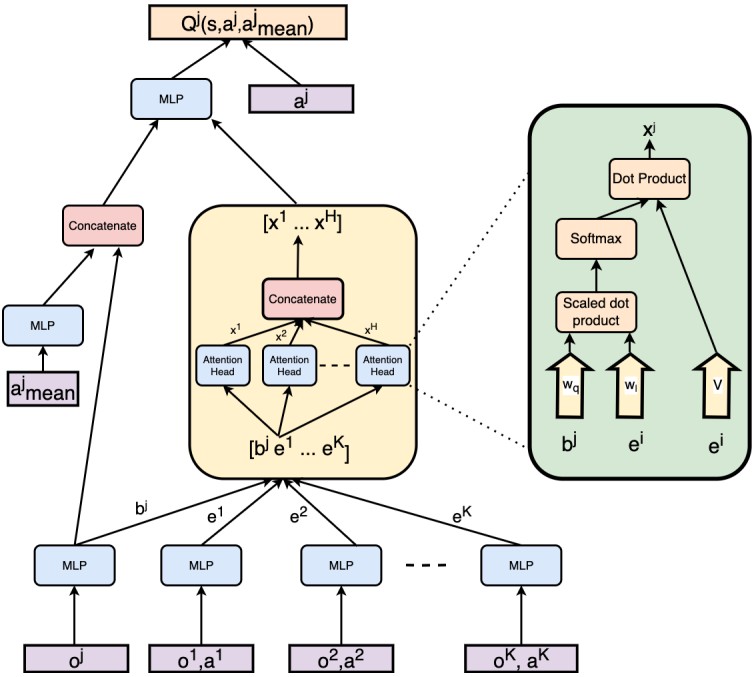

Figure 1: Function approximation architecture for $Q_\phi^j(s, a^j, \overline{a}^j)$ of each agent $j$, where $a_{mean}^j = \overline{a}^j$.

driving and UAV routing, agents move freely and interaction structure changes continuously, i.e., there is no fixed dependence graph, and the influence of distant agents cannot be assumed negligible a priori. The mean field approximation outside the importance set is therefore the principled approach for accounting for these agents in a computationally tractable manner.

MFA enables the central agent to weight different agents in the importance set based on their relative importance. Towards this objective, we incorporate an attention layer (Vaswani et al., 2017) as part of the neural network that serves as the function approximator for the $Q$-function in Eq. 1. The complete architecture is depicted in Figure 1. Let the central agent be indexed by $j$. The inputs to the network are the local observation of the agent $j$, the current observations and actions of other agents in the importance set (assumed to be accessible through the global state $s$) and the mean field action for $j$ (i.e., $\overline{a}^j$ approximates agents outside the importance set). Formally,

$$\overline{a}_t^j = \frac{1}{N - K^j} \sum_{k \notin \mathcal{I}^j} a_t^k \tag{5}$$

Function approximation for the mean field $Q$ function of $j$ considers the agents' own observation, action, and mean field, in addition to having another component that reflects the contributions of agents in the importance set, expressed as,

$$Q_\phi^j(s, \overline{a}^j) = f^j(g^j(o^j), h^j(\overline{a}^j), x^j), \tag{6}$$

where $Q_\phi^j(s, \overline{a}^j)$ is a vector that contains the $Q$-values of all the actions in the action space of agent $j$. From this, the value of $Q_\phi^j(s, \overline{a}^j, a^j)$ can be obtained using the action $a^j$. $\phi$ denotes a set of features. Further, $f^j$ is a multi-layer perceptron (MLP) of three layers, where $g^j$ and $h^j$ are three-layer MLP embeddings. The $x^j$ captures the contribution from other agents in the importance set, expressed as, $x^j = \sum_i \alpha^i v^i = \sum_i \alpha^i n(V^j l^i(o^i, a^i))$, where $l^i$ is an embedding function for the observation and action of the agent $i$, which is linearly transformed by the value vector $V^j$. Also, $n$ is a non-linear activation function. Hence, $v^i$ represents an embedding of agent $i$. $\alpha^i$ is the attention weight that compares two embeddings, one associated with agent $j$'s own observation, i.e., $b^j = g^j(o^j)$, and the other associated with an agent $i$ in the importance set, i.e., $e^i = l^i(o^i, a^i)$, to calculate a

---

**Algorithm 1** Mean Field Attention $Q$-Learning (MFA-QL)

---

1: Initialize the $Q$ functions $Q_{\phi^j}, Q_{\phi^j_-}$, for all agents $j$ configured according to Eq. 6
2: Initialize the mean action $\overline{a}^j$ for each agent $j \in \{1, \dots, N\}$ and the size of importance set $K^j$
3: Initialize the total steps (T), total episodes (E), and initial state $s$
4: **while** Episode $<$ E **do**
5:   **while** Step $<$ T **do**
6:     **while** $j = 1$ to $N$ **do**
7:       Choose action $a^j$ from $Q_{\phi^j}$ according to Eq. 4, using the mean action $\overline{a}^j$, state $s$, and exploration rate $\beta$
8:       Compute the new mean action $\overline{a}^j$ according to Eq. 5 for agents outside the importance set of $j$
9:     **end while**
10:     Execute the joint action $\boldsymbol{a} = [a^1, \dots, a^N]$. Observe $\boldsymbol{r} = [r^1, \dots, r^N]$ and next state $s'$
11:     Store $\langle s, \boldsymbol{a}, \boldsymbol{r}, \boldsymbol{k}, s', \overline{\boldsymbol{a}} \rangle$ in replay buffer $D$, where $\overline{\boldsymbol{a}} = [\overline{a}^1, \dots, \overline{a}^N]$ is the joint mean action, and $\boldsymbol{k} = [k^1, \dots, k^N]$, $\mathcal{I}^j = [o^1, a^1, \dots, o^{K^j}, a^{K^j}]$ denotes the importance set of $j$
12:     Assign $s \leftarrow s'$
13:   **end while**
14:   **while** $j = 1$ to $N$ **do**
15:     Sample a minibatch of $M$ experiences $\langle s, \boldsymbol{a}, \boldsymbol{r}, \boldsymbol{k}, s', \overline{\boldsymbol{a}} \rangle$ from $D$ and store it in $\mathcal{D}$
16:     Using $\mathcal{D}$, set $y^j = r^j + \gamma v_{\phi^j_-}^{MF}(s')$ according to Eq. 1
17:     Using $\mathcal{D}$, update the $Q$ network by minimizing $L(\phi^j) = \frac{1}{M} \sum (y^j - Q_{\phi^j}(s, a^j, \overline{a}^j))^2$
18:   **end while**
19:   Update the parameters of the target network for each agent $j$ with learning rate $\tau$:

$$\phi^j_- \leftarrow \tau\phi^j + (1 - \tau)\phi^j_-$$

20: **end while**

---

similarity value. This similarity value is then passed into a softmax as done in Vaswani et al. (2017). Let $W_q^j$ and $W_k^j$ represent embeddings that transform $b^j$ into a "query" and $e^i$ into a "key" respectively, for the agent $j$. Then the attention weights can be expressed as (where $\intercal$ denotes the transpose), $\alpha^i \propto \exp(e^{i,\intercal} W_k^\intercal W_q b^j)$. We use a multi-head attention in our implementations, where the network has a total of $H$ heads. The objective for each head is to focus on a different weighted mixture of other agents in the importance set. Each head uses a separate set of parameters $(W_k^j, W_q^j, V^j)$, and the final output from each head is concatenated as a single vector.

Finally, by combining all the components discussed so far, we obtain the function approximation described in Eq. 6. Using the update equations provided in Eqs. 1 – 4, along with the attention mechanism, we provide a $Q$-learning algorithm called Mean Field Attention $Q$-Learning (MFA-QL). The complete pseudocode for MFA-QL is provided in Algorithm 1. Here, we use a separate target network for training along with a replay buffer (Mnih et al., 2015). Further, we extend MFA-QL to an actor-critic method called Mean Field Attention Actor-Critic (MFA-AC). In MFA-AC, the $Q$-function is used as the critic and the policy derived from the $Q$-value serves as the actor (details in Appendix H). In Appendix G we prove that theoretical guarentees provided in Yang et al. (2018) for MFRL algorithms also extends to MFA.

Although MFRL methods scale to environments with many agents, they are sample inefficient since they learn policies from scratch (Yardim et al., 2024). To address this, we combine MFA with the algorithm from Subramanian et al. (2022) to accelerate training using pre-existing sources of knowledge by action advising (Silva & Costa, 2019). In Appendix I, we provide the details of this mean field attention advising (MFAA) framework, that combines MFA with action advising, along with an algorithm (MFAA-QL).

Further, using the ideas in Subramanian et al. (2020), we extend MFA to environments having heterogeneous agents based on type classification. The details are in Appendix J. We provide two algorithms using this multi-type mean field attention (MTMFA) framework, MTMFA-QL and MTMFA-AC, extending MFA-QL and MFA-AC to heterogeneous environments.

## 5 Experiments and Results

Experimental results show that the MFA algorithms consistently outperform a set of strong MARL baselines in multiple test beds, and that they maintain their performance advantages even when the number of agents in the environment more than doubles, scaling well to environments with many agents. Specifically, this section provides the following insights:

1. Independent learning algorithms fail to learn well in most many agent environments.

2. Purely mean field based methods lose out when attention to local opponents is important.

3. Purely attention based methods do not scale well to environments with many agents.

4. Methods that combine attention and mean field but ignore agents (outside the neighbourhod) lose out to other methods that do not ignore agents.

5. MFA based algorithms that combines attention, mean field, and does not ignore any agents provide the best performances across a diversity of environments.

6. Action advising helps to provide a warm start for MFA methods leading to noticeable performance advantages.

The most important elements of our experimental domains and implementation details are provided here, while the complete details are in Appendix D. We conduct five sets of experiments (Experiment 1 – Experiment 5) across three many agent test beds: MAgent (Zheng et al., 2017), Neural MMO (Suarez et al., 2021), and SMARTS (Zhou et al., 2021). Similar to prior work in mean field learning (Guo et al., 2019; Subramanian et al., 2020; Yang et al., 2018), we run experiments in two phases — training and execution. All environments contain agents belonging to teams. In the training phase, every agent of the same team trains using the same network and algorithm, consistent with prior work (Subramanian et al., 2020; Yang et al., 2018). After training, agents enter into an execution phase where the trained policies are executed without additional training. We consider eight algorithms: independent $Q$-learning (IL) (Tan, 1993) using neural networks for function approximation (as introduced in Mnih et al. (2015)), mean field $Q$-learning (MFQ) (Yang et al., 2018), mean field actor-critic (MFAC) (Yang et al., 2018), multi-actor-attention-critic (MAAC) (Iqbal & Sha, 2019), weighted-MFRL (WMFRL) (Wang et al., 2022), MFA-QL, MFA-AC, and MFAA-QL. When benchmarking all existing algorithms that use an attention mechanism in MFRL, we found that WMFRL provided the best performance in all of our test beds. We therefore focus on WMFRL here, while extensive comparisons with other algorithms are in Appendix C. Recall that MAAC uses the attention mechanism to attend to every agent in the environment and does not use mean field approximation, while MFQ and MFAC use only the mean field approximation. In our experiments, we consider the entire environment to be the neighbourhood for MFQ and MFAC (our preliminary experiments showed this resulted in the best performance for these two algorithms). During the training phase, MFAA-QL accelerates training using action advising. The advising comes from a pre-trained IL network (trained for 100000 episodes). There is no influence from advisors during execution. We repeat all experiments 30 times and report the average and standard deviation of performances. For training, we report cumulative rewards to track learning progress. For execution in non-cooperative games, we report win percentages in a face-off against MFAA-QL or MTMFAA-QL (the strongest training algorithm), providing a direct measure of policy quality against a strong opponent executing a fixed policy. For execution in cooperative games, we plot the performances of the trained team policy, providing comparisons between the performances of our algorithms and the baselines.

For training analysis, we use a two-sided Wilcoxon signed rank test (typically on the last episode of training), and for execution analysis we use a Fischer's exact test. We report the $p$-values for these tests. Source code for the experiments has been open-sourced (`https://github.com/Sriram94/MFA`).

**Experiment 1**: First, we use the mixed cooperative-competitive Battle game from MAgents (Zheng et al., 2017). This domain contains two teams of homogeneous agents. The game starts with the same number of agents in each team. Each agent is expected to cooperate with other agents of the same team and compete

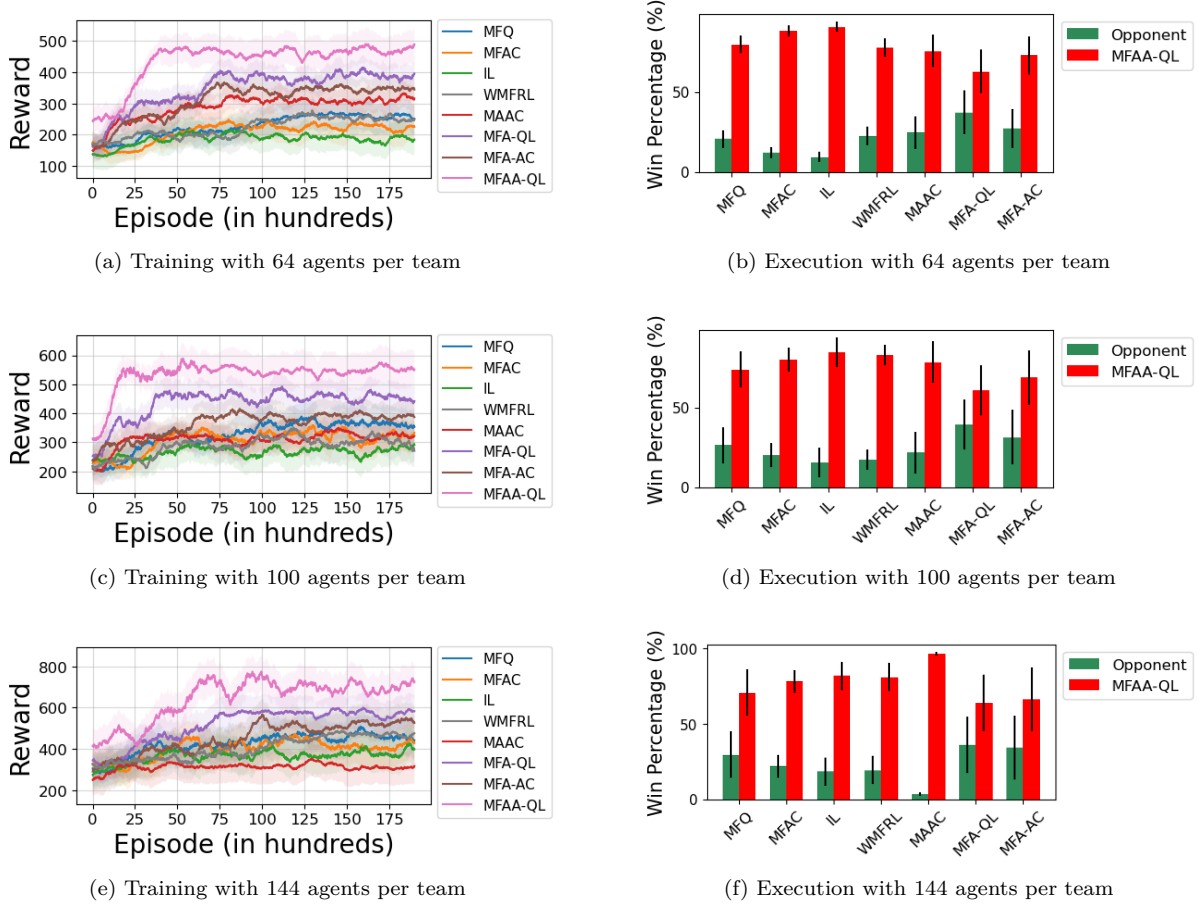

(a) Training with 64 agents per team

(b) Execution with 64 agents per team

(c) Training with 100 agents per team

(d) Execution with 100 agents per team

(e) Training with 144 agents per team

(f) Execution with 144 agents per team

Figure 2: MAgent Battle environment results.

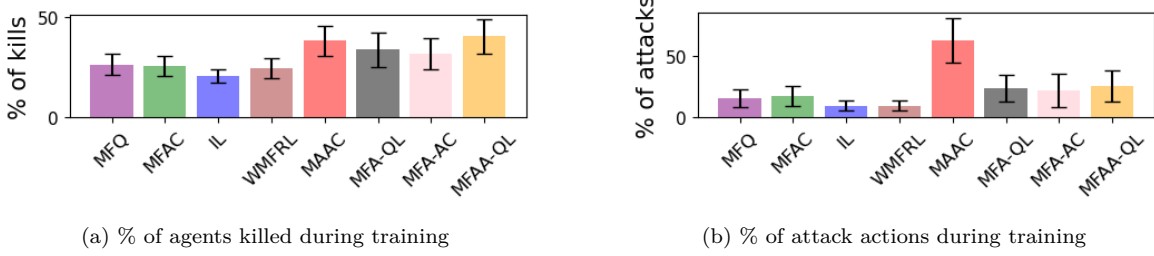

(a) % of agents killed during training

(b) % of attack actions during training

Figure 3: MAgent Battle environment training with 100 agents in each team.

against the agents of the other team to win the Battle. We consider three settings, with each team containing 64, 100, and 144 agents. For training, we plot the cumulative rewards obtained by all agents belonging to one team (the performance of the other team is also similar, our domains are not zero-sum). For execution, we plot the percentage of games won by each algorithm in a face-off with MFAA-QL (the best performing algorithm during training). The team that kills more of its opponents wins the Battle.

Figure 2(a) plots the training performance in the Battle game setting with 64 agents in each team. We make several observations. First, we note that IL which does not use either the mean field or the attention mechanism loses out to all other algorithms. Comparatively, WMFRL uses attention to focus on nearby agents (computing a weighted mean field), but ignores agents further away. Hence, WMFRL performs better

than IL ($p < 0.04$), but loses out to other methods that do not ignore any agents. The mean field-based methods that do not use attention (MFQ and MFAC), lose out to an algorithm that does not use mean field but uses attention across all the agents in the environment (MAAC) ($p < 0.04$). In this game, the mean field can help in longer-term strategic planning but an agent must escape and attack enemies in its immediate vicinity to succeed — attention to local opponents is critical. However, MAAC attends to every other agent in the environment, which includes a lot of noise (agents further away may not pose any immediate threat, and are unnecessary in the attention mechanism). This demonstrates that attending to too many agents makes it harder to really focus on the right agents. Hence, MAAC loses out to MFA-QL ($p < 0.03$), which uses an attention mechanism to attend to nearby agents (a small number of agents in the importance set) and uses the mean field approximation for agents further away. MFA-QL is outperformed by MFAA-QL ($p < 0.01$), which is capable of accelerating training using action advising. We also notice that MFA-QL obtains a better performance than MFA-AC ($p < 0.05$). Prior work notes that $Q$-learning based approaches induce a positive bias through the max action and often outperform actor-critic methods (Yang et al., 2018), which explains the better performance of MFA-QL. Since MFAA-QL uses an attention mechanism, learns from an advisor, and also considers the mean field, it provides better performance than other baselines ($p < 0.01$). We note that action advising provides a warm start that improves sample efficiency. Learning from a comparatively weak advisor (i.e., IL) is better than learning from scratch where the agent needs to only rely on a randomly initialized policy. Supplementary results in Appendix B.2 demonstrates that better advisors indeed lead to experiencing more gains in sample efficiency. The execution results in Figure 2(b) shows that MFAA-QL outperforms others ($p < 0.01$), demonstrating its superiority. Recall, these execution performances in Figure 2(b) are direct head-to-head comparisons of the trained policies of each algorithm against the strongest algorithm from training (i.e., MFAA-QL). Hence, all algorithms demonstrate a comparitively poor performance, though the relative standing is consistent with that seen during training (i.e., MFA-QL provides the most competitive performance to MFAA-QL, followed by MFA-AC, MAAC, and the other algorithms). Note that MFAA-QL vs. MFAA-QL is not shown because this win rate would be 50%, by definition.

Figure 2(c) and (d) shows performances in a Battle setting with 100 agents per team. Again, we note that MFAA-QL provides the best performance in both training and execution ($p < 0.01$). We draw similar inferences for the performances of all other algorithms as in Figure 2(a). However, one notable observation is that MAAC is outperformed by MFQ ($p < 0.05$). This demonstrates that, 1) mean field-based methods are most well-suited to scaling and 2) purely attention-based methods that consider all other agents in the attention mechanism do not scale well. To further analyze the performance of MAAC, we plot the percentage of attack actions executed by each method and the percentage of opponents killed by each method during training in Figure 3(a) and (b). Figure 3(a) shows that MAAC kills almost the same number of opponents on average as MFAA-QL (which is more than MFQ), however, it executes a large percentage of attack actions as seen in Figure 3(b) (much larger than any of the other methods). In the Battle game, agents are penalized for making wrong attack moves, like attacking empty grids or attacking within the same team (details in Appendix D). This leads to MAAC performance being poor compared to other methods as seen in Figure 2(c). We infer that MAAC learns a very aggressive and a relatively naive strategy of attacking the nearest agent (and not seeking opponents to attack), while purely mean field based methods (MFQ, MFAC) are more conservative (very few attack actions). The methods that combine both attention and mean field (MFAA-QL, MFA-QL, MFA-AC) seem to balance aggressive and defensive strategies which is the best approach to win in Battle. This is reflected in their superior performances in Figure 2(c) and (d). In the third Battle setting, with 144 agents per team (see Figure 2(d) and (e)), we note that MAAC performance suffers further and overall even achieves a lower performance than independent IL. MFAA-QL continues to provide the best performance in both training and execution ($p < 0.01$).

**Experiment 2**: For the next set of experiments, we consider anther MAgent environment, the fully cooperative Tiger game. This environment contains a team of tigers that need to work together to kill the deer in the environment. The deer are part of the environment (moving randomly) while tigers are the learning agents. The tigers are rewarded for attacking and killing deer while being penalized for making wrong attacks (attacking other tigers or an empty grid). At least two tigers need to attack a deer together to obtain rewards. We consider two settings with this domain, the first has 54 tigers and 273 deer, and the second has 100 tigers and 500 deer. In the training phase, we plot the cumulative rewards obtained by the team of tigers using each of the seven algorithms. In the execution phase, we plot the average number of

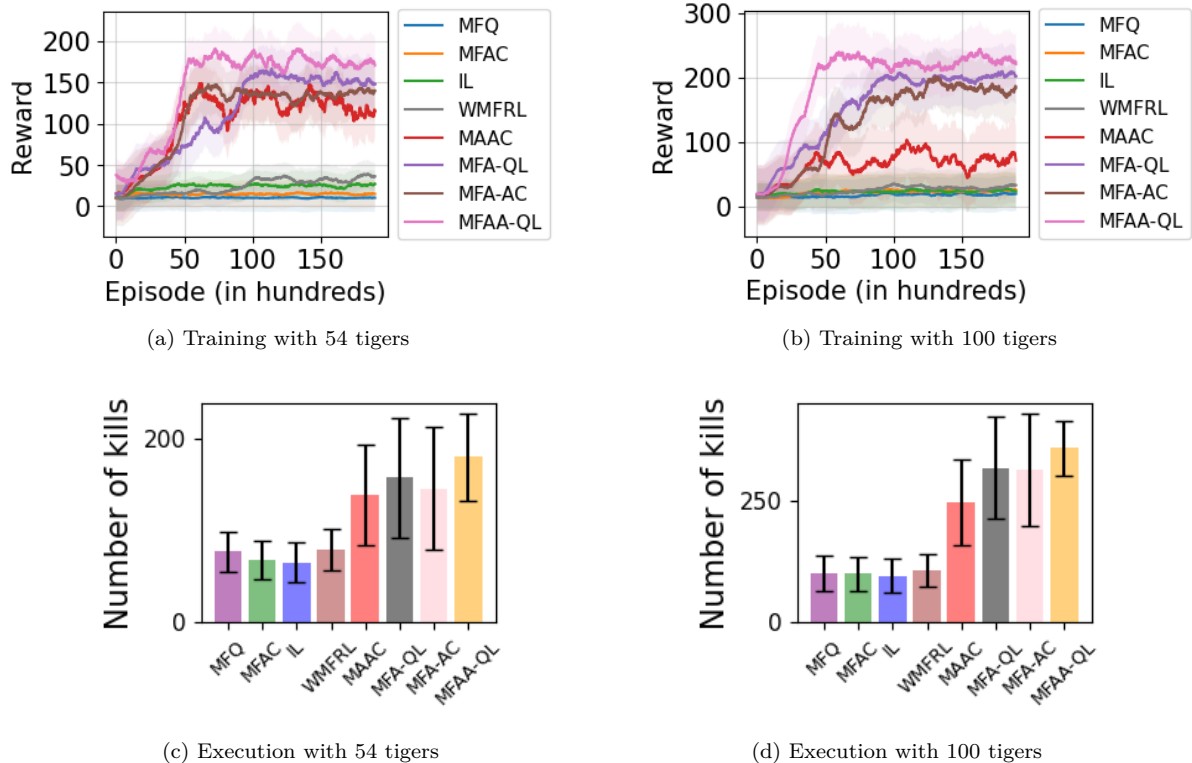

(a) Training with 54 tigers

(b) Training with 100 tigers

(c) Execution with 54 tigers

(d) Execution with 100 tigers

Figure 4: MAgent Tiger environment results.

deers killed (per episode) by the trained policies pertaining to each of the algorithms (no further training). Similar to the Battle game, MFAA-QL uses IL trained for 100000 episodes as the advisor.

First, we consider the setting with 54 tigers. The training results in Figure 4(a) show that the purely mean field methods (MFQ, MFAC) and independent method (IL) lose out completely in this game. This shows that the mean field is not sufficient for learning in this game (since learning to cooperate with nearby agents is required to gain any rewards, local information is more important than global information). The algorithms that use the attention mechanism (MFAA-QL, MFA-AC, MFA-QL, MAAC) show good learning behavior in this domain, with MFAA-QL showing the best performance in training ($p < 0.03$). MFA-QL outperforms MAAC ($p < 0.03$). MFAA-QL provides the best performance in the execution phase (Figure 4c), outperforming all baselines (MAAC, $p < 0.05$) and MFA-QL ($p < 0.8$). Scaling the same setting to 100 tiger agents, we plot the training and execution performance in Figure 4(b) and Figure 4(d) respectively. Most of our observations are the same as the setting with 54 tiger agents (purely mean field methods fail and attention-based methods perform better). MFAA-QL performs the best in both training ($p < 0.05$) and execution ($p < 0.03$). In addition, we note that the performance gap between that of MAAC and MFA-QL widens as compared to the domain with 54 tigers. Again, we note that MAAC does not scale well to an environment with hundreds of agents.

**Experiment 3**: For our third set of experiments, we consider the MAgent Combined Arms heterogeneous environment. The domain is similar to Battle, where we have two teams of agents trying to win by killing members of the opponent team. However, unlike Battle, this environment consists of teams of heterogeneous agents. In each team, there are two types of agents, ranged and melee, with different action spaces. Ranged and melee agents have different abilities and objectives. The ranged agents are faster and can attack further, but lack stamina (i.e., can be killed quickly). The melee agents are slower but have higher stamina (i.e., harder to kill), details are in Appendix D. In this experiment, we model the ranged and melee as distinct types and use the multi-type version of our algorithms (MTMFA-QL and MTMFA-AC), with each team training

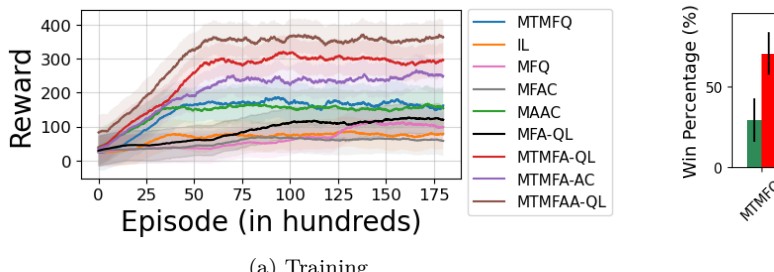
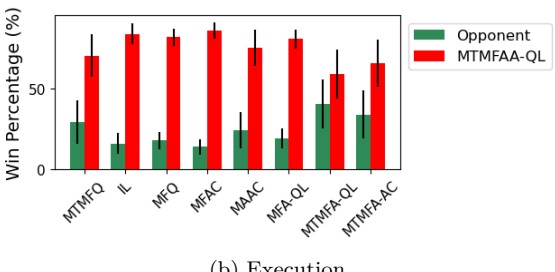

(a) Training

(b) Execution

Figure 5: MAgent Combined Arms Heterogeneous (100 agents per team with two types of 50 agents each) environment results.

a network per type with the same algorithm. We also consider MTMFAA-QL, which accelerates training using action advising. Additionally, MFQ, MFAC, and MFA-QL, algorithms that assume fully homogeneous agents are used as baselines. We are doing this to demonstrate the advantages of type classification for heterogeneous agent environments. To implement these methods we use the action space of the type that contains more actions (i.e., ranged) for the mean field computation (for example, as required in Eq. 3). As mentioned in Section 3, type classification in mean field reinforcement learning was previously studied by Subramanian et al. (2020), and we consider MTMFQ from Subramanian et al. (2020) as a baseline for this experiment. MTMFQ improves MFQ by incorporating types, but does not use any attention mechanism for attending to nearby agents as done in MFA algorithms. Similar to the previous experiments, the advisor for MTMFAA-QL is a pre-trained IL agent (trained for 100000 episodes). Here, each team has a total of 100 agents (50 ranged and 50 melee). We perform training and execution experiments for all algorithms. For execution, we plot the percentage of games won by each algorithm in a face-off contest with MTMFAA-QL (the best performing algorithm during training). The team that kills more of its opponents is determined to win the Combined Arms contest. All other conditions of training and execution are the same as in Battle.

Training results in Figure 5(a) show that MTMFAA-QL obtains the best performance across all algorithms ($p < 0.02$). Further, the other MFA methods that do not use advisors (MTMFA-QL and MTMFA-AC) comfortably outperform all non-MFA baselines ($p < 0.01$). Execution results in Figure 5(b) also show that MTMFAA-QL outperforms others ($p < 0.04$). This is consistent with our observations in Battle demonstrating the importance of the attention mechanism, action advising and mean field computations. Notably, we also demonstrate the importance of type classification for heterogeneous agent environments using this experiment. Algorithms that do not consider types and assume homogeneous agents (MFQ, MFAC, and MFA-QL) perform very poorly in this experiment. This is consistent with observations in Subramanian et al. (2020), which demonstrated the poor performances of MFQ and MFAC across a variety of heterogeneous many agent environments. While MFA-QL outperforms MFQ in this experiment as well, its performance is inferior to MTMFQ ($p < 0.05$) which considers types but does not use the attention mechanism. This shows that for heterogeneous agent environments incorporating types is more important than using the attention mechanism to attend to nearby agents, serving as a motivation for type-based MFA algorithms (MTMFA-QL, MTMFA-AC, and MTMFAA-QL) provided in this work.

Overall, this experiment shows that MFA can be extended to environments with heterogeneous agents and still provide the best performance as compared to previous approaches.

**Experiment 4**: Next, we use the Neural MMO simulator (Suarez et al., 2021) to design a Battle game similar to the MAgent Battle, with 40 agents in each team. The important difference is that the per-agent complexity is much greater in Neural MMO as compared to MAgent (details in Appendix D). We are performing this experiment to test the generalizability of our conclusions, i.e., we want to show that our methods work well on different (possibly harder) platforms as well. For this experiment, we omit MFAA-QL (since MFAA-QL always performs at-least as good as MFA-QL, more discussion in Appendix B). All other conditions of training and execution are the same as in MAgent Battle. Results in Figure 6(a) and (b) show that MFA-QL has the best performance in both training ($p < 0.01$) and execution ($p < 0.04$).

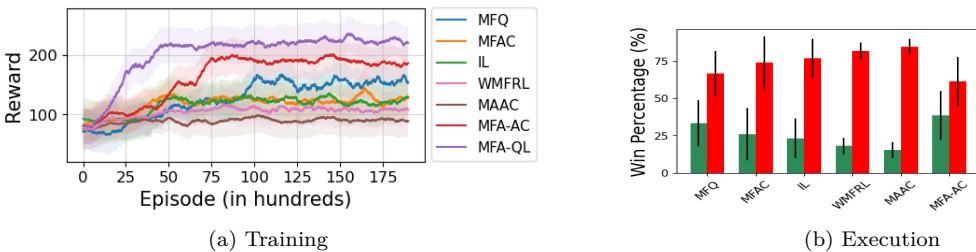

(a) Training

(b) Execution

Figure 6: Neural MMO Battle (40 agents per team) environment results.

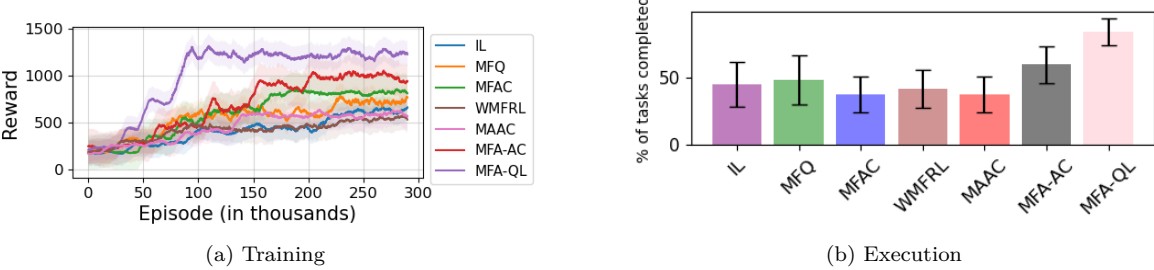

(a) Training

(b) Execution

Figure 7: SMARTS Driving (50 agents) environment results.

**Experiment 5**: Our final domain uses the SMARTS autonomous driving environment (Zhou et al., 2021). We use a set of 50 vehicles (agents) driving on a highway, with each agent tasked with a destination. Agents are rewarded for reaching their destination as fast as possible without any collisions with other vehicles. To reach the destination, agents need to complete a number of tasks (e.g., turning at intersections, slowing down, cruising, and merging). Agents are trained on a diverse set of road and destination configurations (changed every episode). The trained algorithms are then evaluated on a set of previously unseen configurations. Results in Figure 7(a) and (b) shows that MFA-QL outperforms all other algorithms in both training ($p < 0.01$) and execution ($p < 0.03$). In this domain, MFA-QL that formulates global responses (determine speeds, navigation routes, etc.) based on the mean field and local responses (e.g., crash avoidance) based on the attention mechanism shows a clear advantage over all the baseline algorithms which are unable to provide an effective global response and/or an effective local response.

From all the experimental results, we conclude that the mean field has the advantage of scaling to environments with more agents, but loses out when reasoning about individual agents nearby is critical. In contrast, the attention mechanism has the advantage of explicitly reasoning about other agents, but cannot scale easily. Our algorithms combine the best of both worlds, showing the best performance across a diversity of environments. We would like to highlight that, in MFA, distant agents contribute only through the mean field term, which aggregates actions over the entire population. Averaging across many agents reduces variance (via the law of large numbers), providing a smooth and stable global signal rather than noisy individual influences. Furthermore, strong local effects remain captured by the attention module in MFA. Empirically, this combination improves scalability and learning stability in large populations (as demonstrated in the experiments). The *p*-values confirm that our observations are statistically significant. The effect of the importance set and action advising is analyzed further in Appendix B. In Appendix K we provide additional insights on incorporating action advising in a set of baseline algorithms from prior works. In Appendix F we perform a detailed comparison of the wall clock times of all algorithms.

## 6 Conclusion

In this work, we introduced a new mean field paradigm, MFA, and provided two practical algorithms, one based on $Q$-learning (MFA-QL) and the other based on actor-critic (MFA-AC). MFA relaxes restrictive

neighbourhood assumptions in MFRL, where all agents within a pre-defined neighbourhood are assumed to have the same impact and agents outside the neighbourhood have no impact. We showed that MFA training can be accelerated using action advising (Subramanian et al., 2022) and extended to heterogeneous environments using type classification (Subramanian et al., 2020). Further, through extensive empirical investigations, we demonstrated that our algorithms outperform prior approaches in many-agent environments and that scaling the number of agents does not affect its performance. One limitation of our work is that MFA is most effective in settings where a meaningful notion of local proximity between agents is available; domains lacking a natural distance metric or local interaction structure may require learned metrics (see Appendix A for a full discussion of limitations).

## Acknowledgements

Subramanian acknowledges the support from the Canada Research Chairs and the Discovery Grant program from the Natural Sciences and Engineering Research Council of Canada. Poupart was supported by a discovery grant from the Natural Sciences and Engineering Council of Canada, a Canada CIFAR AI Chair and a grant from IITP & MSIT of Korea (No. RS-2024-00457882, AI Research Hub Project). Computational resources used in preparing this research were provided, in part, by the Province of Ontario, the Government of Canada through CIFAR, and companies sponsoring the Vector Institute (`https://vectorinstitute.ai/about/current-partners/`). Part of this work has taken place in the Intelligent Robot Learning (IRL) Lab at the University of Alberta, which is supported in part by research grants from the Alberta Machine Intelligence Institute (Amii); a Canada CIFAR AI Chair, Amii; Digital Research Alliance of Canada; and the National Science and Engineering Research Council (NSERC).

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

## A    Limitations and Future Work

We list a set of limitations of our work and recommendations for future work.

1. Our work assumes the availability of a distance metric between agents for constructing the importance set. In the environments considered in this paper, physical distance between agents serves as a natural and directly observable metric, consistent with prior works that employ locality assumptions between agents (Qu et al., 2022; Yang et al., 2018; Wang et al., 2022; Hao et al., 2023). However, there may exist domains where a natural notion of agent-to-agent distance is not readily available, for example, in purely strategic or abstract settings where agents do not have physical positions. In such settings, the distance metric itself may need to be learned from data (for instance, as in Taylor et al. (2011)), potentially using representation learning techniques. Developing principled approaches for learning agent-to-agent distance metrics in settings where they are not directly observable is an interesting direction for future work, and we expect that combining MFA with such learned metrics would further broaden its applicability.

2. Our work retains the assumption of availability of the global state in MFRL. As prior works have shown, this assumption can be quite strong (He et al., 2021; Malekzadeh & Plataniotis, 2024; Subramanian et al., 2021). Similar to our extension to heterogeneous agent environment, MFA can be readily combined with the work of Subramanian et al. (2021) that extended MFRL to partially observable settings by maintaining an estimate of the uncertainty over the mean field, which is updated using Bayesian updates. We leave this to future work.

3. Our work only applies to discrete action space environments, as is most commonly assumed in the MFRL paradigm (Yang et al., 2018). Extension to continuous action space environments is left to future work.

4. While we demonstrated that MFRL training can be accelerated using action advising, similar gains in sample efficiency are also expected in other forms of multi-agent transfer (Silva & Costa, 2019). We leave a comprehensive analysis of this direction to future work.

5. Recent works have proposed several improvements to the attention mechanism that improves its time and memory complexities. One well-known example is the Flash Attention mechanism (Dao et al., 2022). Further explorations of such architectures in MFRL is left to future work.

6. While our experiments considered one real-world domain of autonomous driving, the MFA can be readily applied to other domains like UAV routing (Wang et al., 2021), several applications in finance (Carmona, 2020), economics (Angiulia et al., 2023), and resource management (Hanif et al., 2015). Extensions to different kinds of real-world applications is left to future work.

## B    Additional Experiments

We provide several supplementary experiments in this section. We conduct statistical significance tests and provide $p$-values consistent with Section 5.

### B.1    Varying the number of agents in the Importance Set

In this sub-section, we perform an experiment using the MAgent Battle domain where we vary the number of agents in the importance set (where $K^j$ represents the number of elements in the importance set for the agent $j$) for the MFAA-QL algorithm (henceforth we will omit the agent index as superscript for the hyperparameter that denotes the importance set $K$, as we use the same value of $K$ for all agents). We consider the Battle game setting with 64 agents in each team. The performances are plotted in Figure 8 where we note that, for a very low value of $K = 4$, the performances are quite poor. This is because a sufficient number of other agents that can impact the performance of the central agent are not being considered in the importance set. When $K$ is increased to 10, we see a large improvement in performance. Increasing $K$ further (20 and 30) shows only a small improvement in performance. In fact, when the value of $K = 40$ the performance drops. This is due to the inclusion of unnecessary information while considering more than a sufficient number of agents (agents further away from the central agent may not have much influence in its decisions). This study shows the importance of choosing the right value of $K$ for the performance of

MFAA-QL. If the value of $K$ is too small, the performances are poor, and if the value of $K$ is too large, there may not be too much gain in performance relative to an increase in computational demands or there may even be a drop in performance. We observed very similar results in each of our other domains as well, so we set the value of $K$ to be 10 throughout.

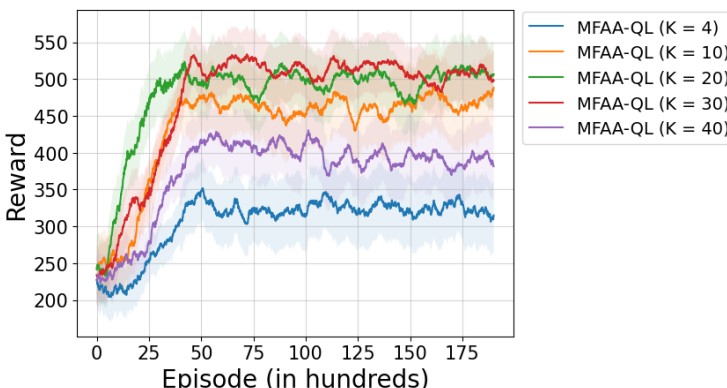

Figure 8: Experiment on MAgent Battle environment varying the value of $K$ (64 agents in each team).

## B.2  Effect of Action Advising

In this sub-section, using the Battle and Tiger domains, we perform an experimental study that considers different types of advisors to analyze the effect of these different advisors. First, we consider the Battle domain with 64 agents in each team (a total of 128 agents). In this domain, we plot the performances of MFAA-QL using 4 different advisors: 1) IL pre-trained for 50000 episodes, 2) IL pre-trained for 100000 episodes, 3) MFA-QL pre-trained for 50000 episodes and 4) MFA-QL pre-trained for 100000 episodes. As noted in our experimental performances in Figure 2(a), MFA-QL performance is far better than IL. Also, within the same algorithm, training for more episodes leads to (at-least) a better performance. Hence, the ranking of these advisors is MFA-QL (100000) > MFA-QL (50000) > IL (100000) > IL (50000), where the number in the brackets denotes the number of pre-training episodes (based on performances, MFA-QL trained for 50000 episodes is better than IL trained for 100000 episodes). The performances of MFAA-QL in Figure 9(a) show that MFA-QL (100000) provides the best performance and IL(50000) provides the least performance. This shows that comparatively better advisors lead to better performances and vice-versa. Next, we conduct the same study (with the same set of advisors) in the Tiger domain with 54 agents. As noted in Figure 4(a), the IL algorithm hardly learns anything in this domain and the MFA-QL method reaches a relatively very stable equilibrium point after 20000 episodes of training. Hence, the order of advisors here are MFA-QL (100000) $\approx$ MFA-QL (50000) > IL (100000) $\approx$ IL (50000). The performances of MFAA-QL in Figure 9(b) show a relatively better performance with better advisors (and vice-versa) as noted in the Battle game.

## C   Additional Baselines

In this section we compare MFA algorithms to a set of baselines in addition to the algorithms used in the main paper (Section 5). Out of all the previous algorithms that used attention mechanism in MFRL, WMFRL (Wang et al., 2022) provided the best performance across all our test beds and hence, we used this method in the main experiments in Section 5. Further, some methods such as GAT-MF (Hao et al., 2023) requires all agents in the environment to have fixed relative positions throughout the game so that adjacency among them can be used to build a static graph. All of our experiments in Section 5 use dynamic games where agents can move freely in space. Hence, we did not consider GAT-MF as a baseline in Section 5. Nonetheless, we provide comparisons of MFA algorithms to GAT-MF in this section.

Further, we did not use centralized algorithms as baselines in our main paper since these algorithms maintain and update a $Q$-function over the joint action space, which does not scale well to environments with large

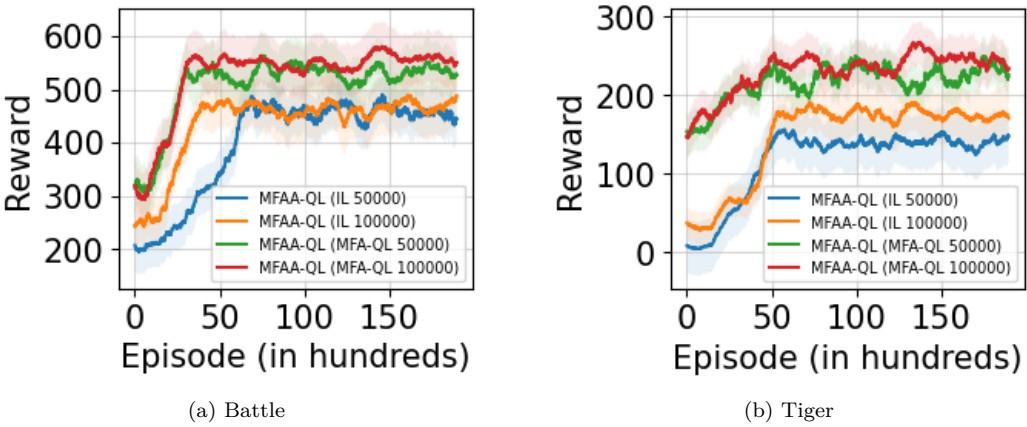

(a) Battle

(b) Tiger

Figure 9: Experiment that shows the influence of advisors on the MAgent Battle and Tiger environments. The Battle domain in this study uses 64 agents in each team while the Tiger domain uses 54 tiger agents.

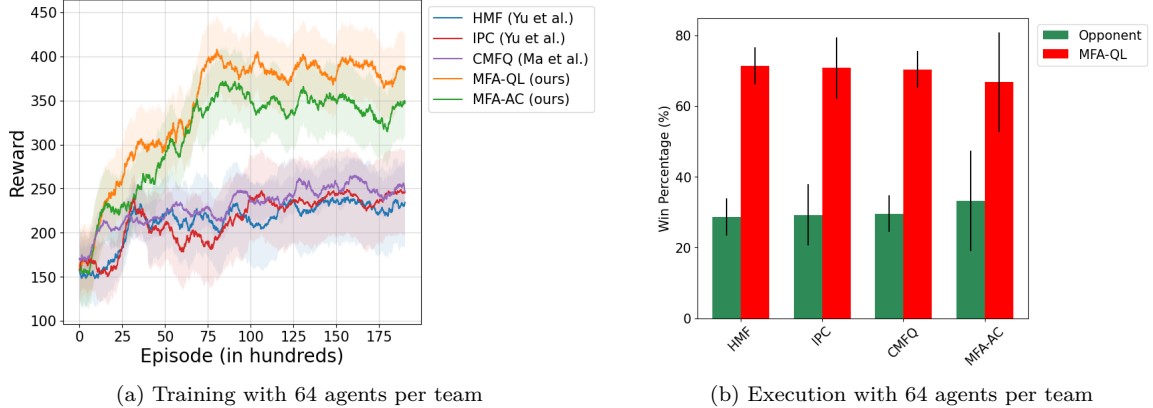

(a) Training with 64 agents per team

(b) Execution with 64 agents per team

Figure 10: Comparisons of MFA algorithms with Yu (2023) and Ma et al. (2023) in the MAgent Battle environment with 64 agents.

numbers of agents. Nonetheless, we provide comparisons to centralized algorithms with MFA algorithms in this section.

In general, independent learning methods did not work well in our experiments since they (rather naively) assume all agents to be just part of the environment. Prior works in many agent learning (Hao et al., 2023; Subramanian et al., 2022; Wang et al., 2022; Yang et al., 2018) also report the same observation. In the main paper, we included comparisons to independent $Q$-learning (IL) that gave the best performances across different independent techniques in our test beds. Nonetheless, we provide more comparisons to other independent techniques in this section.

The AMFQ (Wang et al., 2024) technique is very similar to WMFRL, except for minor changes in the network architectures used. However, we find that WMFRL significantly outperforms AMFQ in all of our experimental domains. As noted in Section 3, the major reason is that WMFRL incorporates a double $Q$-learning update (Hasselt, 2010) (while AMFQ does not) that mitigates the overestimation bias commonly found in $Q$-learning methods. Hence, we do not consider AMFQ as a baseline while we compare extensively to WMFRL in Section 5.

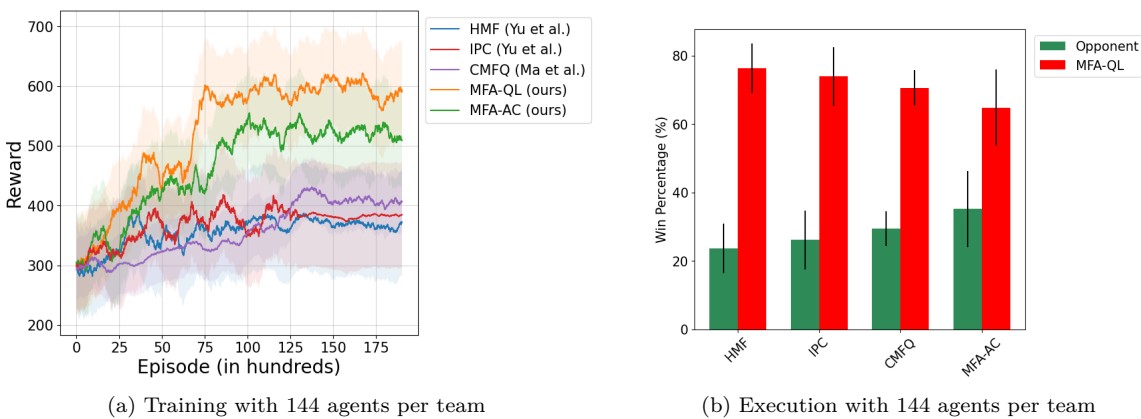

(a) Training with 144 agents per team

(b) Execution with 144 agents per team

Figure 11: Comparisons of MFA algorithms with Yu (2023) and Ma et al. (2023) in the MAgent Battle environment with 144 agents.

## C.1 Comparisons to Hierarchical MFRL and Causal MFRL

Yu (2023) proposes a hierarchical mean field (HMF) framework where all the agents in the environment are divided into a set of groups on the basis of the actions performed by each agent in a finite history of experiences. The mean field approximation is used to approximate the influences within the group on each agent. Further, the influence of each group on the global population is computed using a summation of each group $Q$-function.

Ma et al. (2023) proposes the causal mean field reinforcement learning framework that incorporates causality by using a structural causal model (SCM) within the neighbourhood of agents in a mean field setting. In place of using the attention mechanism for determining the weights of a weighted mean field as done by Wang et al. (2022), Ma et al. (2023) uses the SCM for determining the degree of interactions (i.e., weights) of agents within the neighbourhood. A weight is assigned to each neighbourhood agent that reflects the neighbouring agent's causal effect on the policy of the central agent. Similar to WMFRL, agents outside the neighbourhood are ignored in this framework as well.

We perform comparisons of MFA with the algorithms in Yu (2023) and Ma et al. (2023) on the MAgent Battle game with teams containing 64 agents and 144 agents. All the conditions for training and execution are the same as those described in our paper for the MAgent Battle game used in Section 5. We provide these results for 25 random seeds and report statistical significance as in Section 5. Yu (2023) contains two algorithms (HMF and IPC) both of which are used for these experimental comparisons against our algorithms (MFA-QL and MFA-AC). Further, we use the causal MFQ (CMFQ) algorithm from Ma et al. (2023) for these comparisons.

From the results in Figure 10 and Figure 11, it can be seen that MFA-QL significantly outperforms all three algorithms (HMF, IPC, CMFQ) in all experimental settings ($p < 0.01$ for both training and execution). Regarding CMFQ, since this algorithm ignores agents outside the neighbourhood, its performance suffers, same as our explanations for WMFRL in Section 5. Since the MFA algorithms do not ignore any agents, their performance is superior to CMFQ. Regarding HMF and IPC, there are several reasons for their observed poor performance in these experiments. First, HMF and IPC divides all the agents in the environment into a set of $m$ finite groups and the mean field approximation is applied for each group. Further, a sum of $Q$-values of each group is used to find the global/total $Q$. As shown in our Theorem 1, there is an approximation error for each mean field approximation, which are added together when the sum of $m$ approximated $Q$-functions is computed. Hence, the total approximation error in HMF and IPC is bounded by $mL$ (where $L$ is the Lipshitz constant in Theorem 1), while MFA algorithms perform only one mean field approximation leading to the approximation error being bounded by $L$. Second, HMF and IPC use a local $Q$-function for action selection that does not depend on the mean field (see Step 4 of Algorithm 1 in Yu et al.). While the mean field is used to update the $Q$-functions through a Bellman error, it is not used for action selection which leads to a drop in

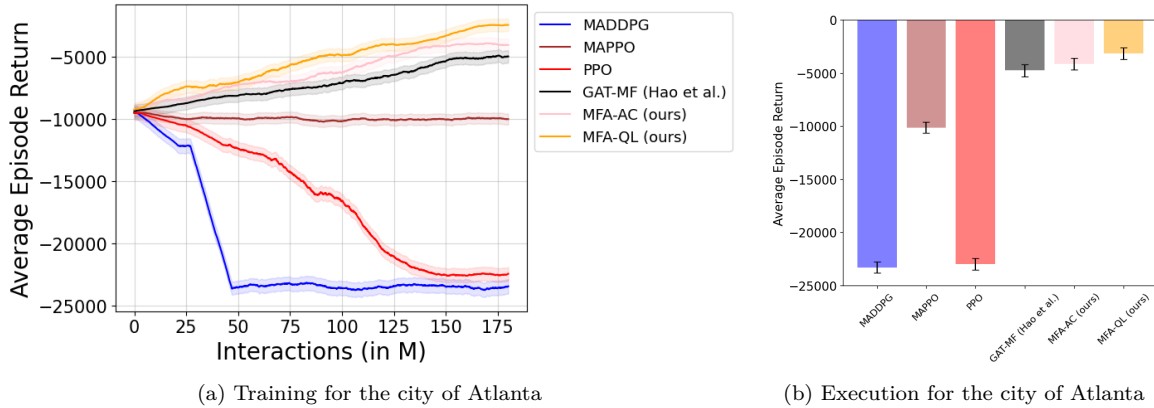

(a) Training for the city of Atlanta

(b) Execution for the city of Atlanta

Figure 12: Comparisons of MFA algorithms with GAT-MF (Hao et al., 2023) in the COVID-19 Vaccination task developed by Hao et al. (2023) in the city of Atlanta. The training plots on the left are over millions of interactions (M).

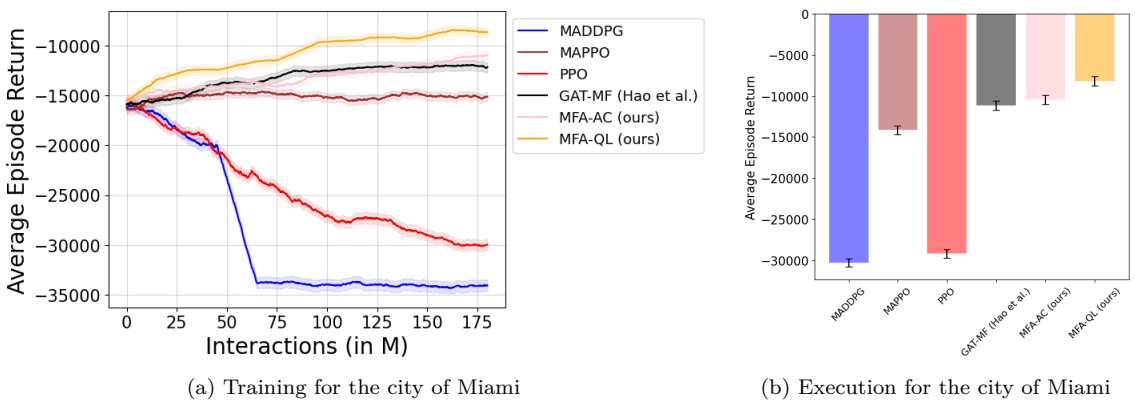

(a) Training for the city of Miami

(b) Execution for the city of Miami

Figure 13: Comparisons of MFA algorithms with GAT-MF (Hao et al., 2023) in the COVID-19 Vaccination task developed by Hao et al. (2023) in the city of Miami. The training plots on the left are over millions of interactions (M).

performance. Third, HMF and IPC performs dynamic assignment of agents to groups based on observation and action similarities of agent experience histories. In dynamic games like Battle where agents need to actively change their strategies quickly, their group allocations become rather arbitrary as past experiences are not a good indication of current behaviours. Fourth, HMF and IPC include a regularization term that constrains the policy updates in such a way that the distance of each agents policy from the average group policy stays bounded. Since the group allocations are rather arbitrary, this regularization heavily constrains the policy updates to some arbitrary range, leading to a loss in performance. Fifth, IPC computes the average historical group policy and includes an addition term that adjusts the policy update of each agent within a group to be similar to historical behaviours of other agents within that group. Again, this leads to poor performance and slow learning in dynamic games like Battle where the policy updates are constrained by rather irrelevant past behaviours in arbitrary groups.

## C.2 Comparisons to GAT-MF

We have performed additional comparisons to the GAT-MF algorithm from Hao et al. (2023) using the same COVID-19 vaccine allocation real-world metropolitan task considered by Hao et al. which contain more than 3000 agents in each task. Please refer to Hao et al. (2023) for the complete details of this domain. We plot

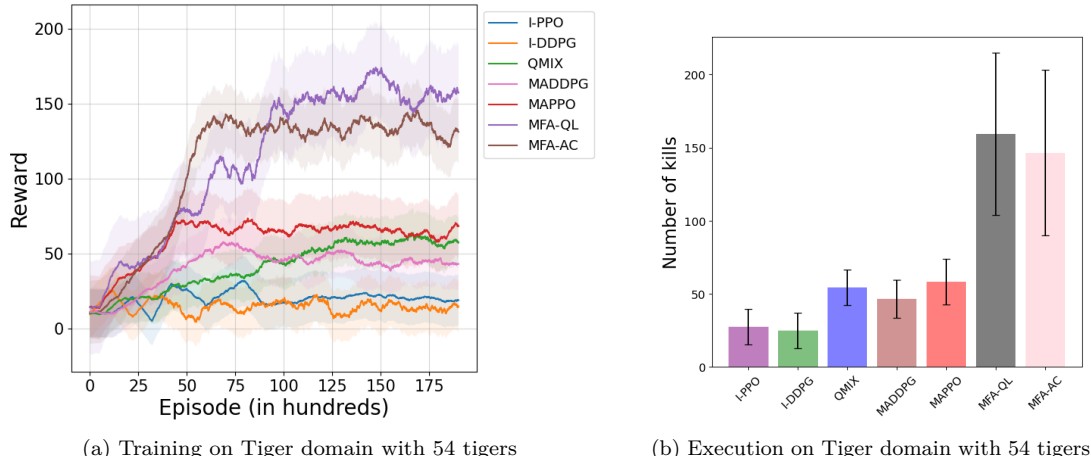

(a) Training on Tiger domain with 54 tigers          (b) Execution on Tiger domain with 54 tigers

Figure 14: Additional baseline comparisons on the Tiger environment.

results for allocations in the city of Atlanta and Miami same as those considered by Hao et al. We conduct the experiment in training and execution phases as in other experiments in our paper. We report training and execution results for 30 random seeds (while Hao et al. consider 5 random seeds) and conduct the Wilcoxon signed rank test for statistical significance. For the experiments, we use a set of centralized training algorithms (MADDPG, MAPPO), independent algorithms (PPO), the method from Hao et al. (GAT-MF), and our algorithms (MFA-QL, MFA-AC).

The results are in Figure 12 for the city of Atlanta and Figure 13 for the city of Miami. Across all the results we see that MFA methods outperform GAT-MF in both training and execution ($p < 0.01$ for training and $p < 0.03$ for execution). GAT-MF uses the Graph Attention Network to compute a weighted mean field across neighbours in the static graph and ignores agents outside this neighbourhood. This is similar to our WMFRL baseline that uses an attention network to compute a weighted mean field in the neighbourhood and ignores agents outside the neighbourhood. As shown across multiple experiments in our paper, ignoring agents outside the neighbourhood leads to a poor performance. MFA-QL does not ignore any agents, it uses the attention mechanism to attend to agents nearby and the mean field approximation for agents further away. Hence it outperforms other algorithms that actively ignore agents (such as GAT-MF and WMFRL).

Further, we see that MFA-QL outperforms algorithms that use centralized training (MADDPG and MAPPO). These algorithms learn over the joint state-action space of all agents which does not scale well to environments with many agents. The rationale for the poor performance of these algorithms are similar to that of MAAC (reasoning about all agents in the environment including agents further away makes it hard to focus on the right agents leading to slow learning and poor performance).

The independent PPO algorithm completely fails to learn in these environments (as also reported by Hao et al. (2023)) and is outperformed by our methods.

## C.3  Comparisons to additional Independent Learning and Centralized Training algorithms

We perform additional experiments comparing the performance of MFA algorithms (MFA-QL, MFA-AC) to a set of additional baselines: Independent PPO (I-PPO), Independent-DDPG (I-DDPG), Global/Multi-agent PPO (MAPPO), Multi-agent DDPG (MADDPG), and QMIX. Since some of these algorithms are restricted to cooperative environments, we perform our experiments on the Tiger cooperative domain used in the paper. We use a 54-agent Tiger environment and a 100-agent Tiger environment, as done in Section 5. We report training and execution results for 25 random seeds along with statistical significance results (using the Wilcoxon signed rank test for training and the Fischer's exact test for execution).

The results for training and execution are in Figure 14 for the 54-agent Tiger environment and Figure 15 for the 100-agent Tiger environment. From the results we see that our methods (MFA-QL, MFA-AC)

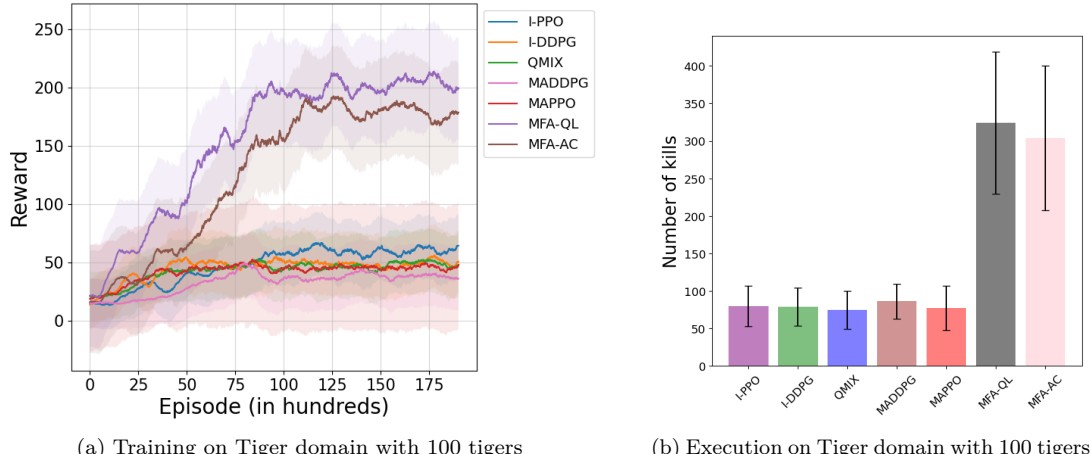

(a) Training on Tiger domain with 100 tigers      (b) Execution on Tiger domain with 100 tigers

Figure 15: Additional baseline comparisons on the Tiger environment.

outperform all these additional baselines across both training and execution in both environments ($p < 0.01$ for all comparisons). The centralized learning algorithms (MAPPO, MADDPG, QMIX) learn over the joint state-action space of all agents which does not scale well to environments with many agents. Furthermore, the rationale for the poor performance of these algorithms are similar to that of MAAC (reasoning about all agents in the environment including agents further away makes it hard to focus on the right agents leading to slow learning and poor performance).

Independent learning algorithms (I-PPO, I-DDPG) fail to learn good policies in these environments and are outperformed by our methods. As seen in multiple experiments in our paper, independent methods generally perform poorly in many agent environments since they are not able to reason over the strategies of other agents. This becomes extremely important in several many-agent environments where agents need to constantly change their strategies and react to other agents nearby.

## D   Experimental Details

In this work, we used three experimental testbeds — MAgents, Neural MMO, and SMARTS. More details about each of these domains are given in this section. For the non-cooperative homogeneous experiments, our training procedure resembles self-play (Shoham et al., 2003) where the different teams train separate networks using the same algorithm, and evaluation is head-to-head for our method vs. each baseline. For heterogeneous environments, the same training scheme is used with two networks, one for each type of agent. For cooperative experiments, all agents train the same network for each algorithm during training, which is then evaluated in a seperate evaluation phase.

### D.1   MAgents

We consider three domains in the MAgents testbed. These are the Battle, Tiger, and the Combined Arms environments.

The Battle domain is a mixed cooperative-competitive domain that contains two teams of agents, where an agent is trying to kill the members of the opponent team by cooperating with the members of the same team. In this game, each agent starts with a maximum horsepower (HP) of 10 units. The HP denotes the energy of the agent. This HP gets depleted upon being attacked by opponents. The HP is gradually replenished over time, and the agent dies when the HP becomes 0. Each agent loses 2 HP upon being attacked by an opponent and is able to recover 0.1 HP at every step. Each agent gets a small penalty of -0.005 at every step, a punishment of -0.1 for dying (dead penalty), and a punishment of -0.01 for a wrong attack (attacking agents of the same team or an empty grid). An agent's attack on another agent in its own team is not registered. The agents gain a reward of +0.5 upon attacking opponents and a reward of +10 upon killing an opponent.

Each episode in this setting constitutes a full battle of a maximum of 1000 action steps (or cycles). The game terminates if either all the steps are done or if one of the two teams completely dies. The observation space of each agent is a $13 \times 13$ map around the agent where the agent gets full observation (and no observation outside this radius). The entire state space is a grid of dimensions ranging from $40 \times 40$ (for environments with 64 agents in each team) to $60 \times 60$ (for environments with 144 agents in each team). Each agent can perform a total of 21 discrete actions, of which one action pertains to doing nothing, 12 actions pertain to moving, and 8 actions pertaining to attacking another agent nearby. The conditions in this game are almost the same as the defaults in the Petting Zoo library (Terry et al., 2020). In the execution experiments with the Battle game, a game is determined to be won by the team that kills more of its opponents. If both teams kill the same number of opponents, the team obtaining higher cumulative rewards is determined to be the winner.

Another domain we consider is the Combined Arms heterogeneous mixed cooperative-competitive environment (Terry et al., 2020). The conditions of this game are similar to the Battle game, except that each team consists of 2 heterogeneous groups of agents — ranged and melee. The ranged agents can move faster and attack further but have a lesser maximum HP of 3 units (can be killed easily as compared to melee agents). The melee agents can only attack other agents very close to themselves but have more energy (more maximum HP of 10 units as compared to ranged agents). The melee agents have a total of 9 actions which consists of doing nothing, 4 attack actions, and 4 move actions. The ranged agents have a total of 21 actions involving doing nothing, 12 move actions, and 12 attack actions. The entire state space is a grid of dimensions $50 \times 50$. All other conditions of this game are the same as the Battle game.

Another MAgent environment we consider is the fully cooperative Tiger game. In this game, the tigers start with a maximum HP of 18 units, where they lose 0.02 HP at every step they do not eat a deer. If they do eat a deer, they gain an HP of 8. When the HP becomes 0, the tiger dies. The deer have a maximum HP of 5, lose 1 HP when attacked, and regain 0.1 HP at every step. So it is better to kill a deer quickly. The tigers have an action space of 9, where there is one action to signify doing nothing, 4 attack actions, and 4 move actions. The action space of the deer consists of 5 actions (including doing nothing and 4 move actions, the deer cannot attack). The tigers receive a reward of +2 for attacking a deer alongside other tiger(s). At least two tigers have to attack a deer together to obtain a reward. The tigers get a penalty of -0.01 in case of a wrong attack (attacking empty grids, other tigers, or deer alone). The state is a grid of dimensions ranging from $74 \times 74$ (environment containing 54 tigers) to $100 \times 100$ (environment containing 100 tigers), with a tiger getting an observation of dimension $9 \times 9$ around itself. Same as the Battle domain, each episode in this setting constitutes a full game of a maximum of 1000 action steps (or cycles). An episode ends either if all the deer are killed or if all the steps are completed.

### D.2 Neural MMO

Neural MMO, originally released by Suarez et al. (2021), is a simulation of the popular massively multiplayer online (MMO) (Statista, 2021) games. This environment supports learning with many concurrent learning agents, as in the MAgent platform. However, the per-agent complexity of agents in Neural MMO is much larger than that of MAgents. We consider a mixed cooperative-competitive setting where we have two groups of agents with 40 agents on each team in a $128 \times 128$ grid (medium configuration from Suarez et al. (2021)). The objective of an agent is the same as the MAgent Battle game, where each agent tries to kill agents belonging to the opponent team by cooperating with agents in the same team. At the start of the game, each agent spawns randomly at any one of the grids with a maximum HP of 100. At each step, an agent loses 20 units of HP upon being attacked and gains 1 HP every step at which it is not attacked. The agent also loses different quantities of HP based on its current stock of food and water resources (see Suarez et al. (2021) for more details). The agent dies when the HP drops to 0 or lower. The observation of each agent contains two components, where the first component contains entity information (i.e., information about the agent itself including previous action and HP) and the second component includes a local view of $15 \times 15$ around an agent. The action space is a discrete set of 108 actions, where 103 actions pertain to attack (including direction and style of attack) and 5 actions pertain to moving (into one of the 4 cardinal directions and doing nothing), see Suarez et al. (2021) for more details. Agents get a reward of +150 upon killing an opponent and +5 for attacking an opponent. The agents lose -5 for dying and -2 for a wrong attack. In addition, agents

also get rewarded for exploration, foraging, and equipment, and the values are the same as the defaults in Suarez et al. (2021).

### D.3 SMARTS

The SMARTS simulator from Zhou et al. (2021) provides several training scenarios replayed through a simulation platform. In this platform several driving scenarios are constructed on the basis of recorded human behaviours from open-source driving datasets (such as NGSIM (Alexiadis, 2006) and Waymo (Sun et al., 2020)). Each scenario consists of negotiating a number of driving tasks such as slowing down, changing lanes, merging, cruising, and overtaking. There are a total of eight such tasks defined in SMARTS (see Zhou et al. (2021) for more details). In our environment, we have a set of 50 driving agents. For each scenario, each agent is tasked with navigating to a destination as fast as possible. This navigation requires learning to successfully complete a number of different combination of driving tasks. Agents are rewarded based on the number of tasks completed in each scenario (using the completion ratio, see Zhou et al. (2021)) and the average time it takes for the agent to complete the entire scenario. The state space for each agent consists of several types of different sensor data. SMARTS provides built-in adapters to convert sensor data to tensors which we use as state information for the agents. The action space is a four-dimensional vector of discrete values, two for longitudinal control (keep lane and slow down) and two for lateral control (turn right and left). Refer to Zhou et al. (2021) for more details about the state and action spaces.

## E Hyperparameter and Implementation Details

Most hyperparameters are the same or closely match prior works in mean field learning (Yang et al., 2018). We make small changes for performance and computational efficiency reasons.

For the WMFRL baseline, we use the $Q$-learning variant from Wang et al. (2022) (denoted as WMFQ in Wang et al.), since we observed that WMFQ provided a better performance than the actor-critic variant (WMFAC), consistent with the observation in Wang et al. (2022).

For the MAgent environment experiments, the $Q$-learning-based methods (IL, MFQ, MFA-QL, WMFRL, MFAA-QL, MTMFA-QL, and MTMFAA-QL) almost use the same hyperparameters, where the learning rate is $\alpha = 10^{-2}$, with an $\epsilon$-greedy exploration rate of 1%. The discount factor $\gamma = 0.9$, and we use a mini-batch size of 64. The replay buffer memory size is $2 \times 10^4$. IL and MFQ use an evaluation and target network that has 4 fully connected layers (3 ReLU layers of 50 neurons and an output layer). MFQ uses separate parallel encoders for state and mean field (as in Yang et al. (2018)), feeding into two hidden layers and the final output layer. The target network is replaced after every 10 learning iterations using the hard replacement strategy. The architecture of MFA-QL and MTMFA-QL are as specified in Section 4 and Appendix J respectively. The temperature for Boltzmann's policy is set to 0.1.

For the MAgent environment experiments, the actor-critic-based algorithms (MFAC, MAAC, MFA-AC) almost use the same hyperparameters. The learning rate is $10^{-2}$ for the critic, $10^{-4}$ for the actor, and the temperature for Boltzmann's policy is 0.1. MFAC uses a critic and actor-network with an architecture of 4 fully connected layers (3 ReLU layers of 50 neurons and a linear output layer). The actor networks of MAAC and MFA-AC also use the same configuration. The critic of MFAA uses the configuration given in Iqbal & Sha (Iqbal & Sha, 2019). The critic of MFA-AC uses the configuration given in Section 4. The target networks are replaced every 10 learning iterations using the hard replacement strategy.

For the Neural MMO and SMARTS experiments, all hyperparameters and network architectures are the same as mentioned for the MAgent experiments. The exceptions are: 1) the neural network configurations used in all algorithms contain one additional fully-connected layer, and 2) all network layers contain 200 neurons.

All algorithms that use the attention mechanism (MAAC, MFA-QL, MFA-AC, MFAA-QL, MTMFA-QL, MTMFA-AC, MTMFAA-QL) use a total of four attention heads (value of $H$) in all of our environments. Leaky ReLU is used as the non-linear activation function in all of these networks involving the attention mechanism.

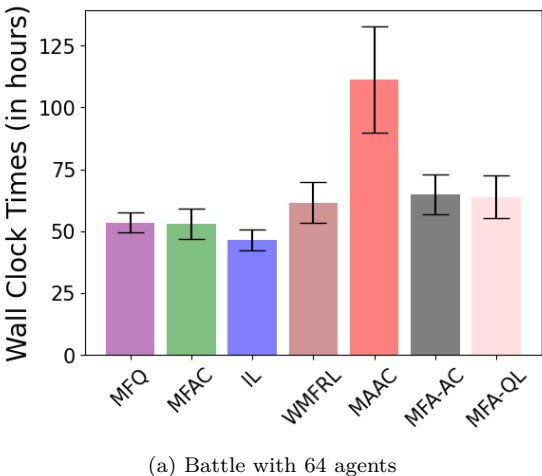
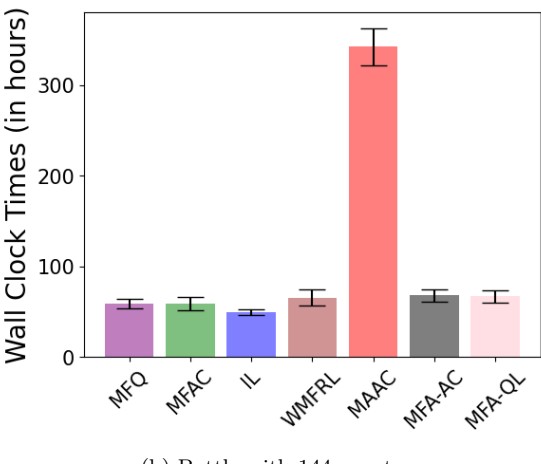

(a) Battle with 64 agents           (b) Battle with 144 agents

Figure 16: Plot of Wall Clock Times in the MAgent Battle environment for all algorithms, with a) 64 agents and b) 144 agents.

The hyperparameter $K^j$ (importance set) in MFA-QL, MFA-AC, and MFAA-QL is set to 10 throughout, i.e., we consider the 10 closest agents (based on physical distances) as part of the importance set for each agent (Appendix B.1 explains the choice of $K^j$). For the MAgent Combined Arms heterogeneous environment, the importance set for MTMFA-QL, MTMFA-AC, and MTMFAA-QL contains the nearest 5 agents of each type (5 ranged and 5 melee) for the same total of 10 agents based on distances to the central agent. We also define a neighbourhood for WMFRL that contains at-least 10 agents to be consistent with the importance set of MFA methods. This provided the best performances for WMFRL in our experiments (similar to Yang et al. (2018), Wang et al. (2022) do not provide any approach to determine the neighbourhood size).

In our experiments, we consider the entire environment to be the neighbourhood for MFQ and MFAC since this gives the best performance (since no agent is ignored), and Yang et al. (2018) did not provide any approach to determine the neighbourhood in a given environment. For algorithms that use advising (such as MFAA-QL and MTMFAA-QL), the hyperparameter $\epsilon'$ starts at 1 at the beginning of training and decays to 0 within 500 episodes of training. There is no influence of advisors for the remaining training episodes. Since our domains support a variable number of agents (including killing), if the number of agents in the environment goes less than $K^j$, we use a placeholder value (all zeroes) for the observation and actions of the remaining agents. This is also done for our implementation of MAAC, since MAAC does not naturally support learning in environments with a variable number of agents.

## F   Wall Clock Times

The majority of training for all the experiments on the MAgent domains was run on a 4 GPU virtual machine with 40 GB GPU memory per GPU. We use Nvidia Ampere-100 (A100) GPUs for all these experiments. The CPUs use AMD Milan 7413 as the microprocessor architecture. We have a CPU memory of 200 GB. The MAgent Battle and Combined Arms experiments took an average of 5 days of wall clock time to complete for all the considered algorithms. The tiger experiments took an average of 3 days wall clock time to complete.

The majority of training for the SMARTS and Neural MMO domain was done on a virtual machine of a similar configuration, except that we use a CPU memory of 300 GB. The algorithms took an average of 7 days wall clock time to complete.

To provide a comparison of wall clock times of different algorithms and their ease of scaling, we provide a plot of the wall clock times taken by each algorithm for training 20000 episodes in the Battle game using the same computational resources (as noted above). We plot the wall clock times for 7 algorithms omitting

MFAA-QL (MFAA-QL takes almost the same amount of wall clock time as MFA-QL, since action advising uses a pre-trained network that does not involve any significant gains in wall clock times).

The plot in Figure 16(a) is for the Battle game setup with 64 agents. From this plot we note that independent techniques (IL) are fastest to train, and this is followed by algorithms that only use the mean field (MFQ, MFAC) without an attention mechanism. Algorithms that use the attention mechanism and the mean field together (WMFRL, MFA-AC, MFA-QL) take more time, but not significantly longer. Given their better training and execution performances (especially for MFA-QL and MFA-AC) as noted in Section 5, it is quite desirable to tradeoff small increases in training times to obtain large improvements in performance. The MAAC method that uses the attention mechanism to attend to all agents in the environment, takes more than twice the wall clock time for training as compared to other algorithms.

The plot in Figure 16(b) shows the wall clock times for the Battle game with 144 agents. The order of wall clock times across algorithms is the same as that observed for the case with 64 agents (independent technique < purely mean field techniques < mean field and attention techniques < purely attention techniques). Comparing the training times between the 64 agent and 144 agent environments, we note that all algorithms except MAAC require only a small increase in training times when the number of agents are more than doubled. However, the wall clock time of MAAC almost triples when the number of agents is doubled. Recall, MAAC also performed poorly (in both training and execution) in the Battle game with 144 agents as seen in Section 5. Thus, MAAC scales very poorly with the number of agents in the environment based on both wall clock times and performances.

## G  Theoretical Results

We provide two theoretical results, one regarding the mean field approximation when locality and homogeneity assumptions are relaxed, and the other regarding convergence.

We first define the two key assumptions used in Theorem 1. Following Yang et al. (2018), we say that the multi-agent Q-function $Q^j(s, a)$ is **additively decomposable** if it can be factorized into pairwise interactions between the central agent $j$ and each other agent $k$, i.e.,

$$Q^j(s, a) = \frac{1}{N} \sum_k Q^j(s, a^j, a^k), \tag{7}$$

where $N$ is the number of agents. Further, we say that $Q^j(s, a)$ is **$L$-smooth** if its gradient is Lipschitz continuous with respect to the actions of other agents, i.e., with respect to $a^k$ for all $k \neq j$, with Lipschitz constant $L$. Specifically, for any two action vectors $a^k$ and $a^{k'}$,

$$\|\nabla_{a^k} Q^j(s, a) - \nabla_{a^{k'}} Q^j(s, a)\| \leq L \|a^k - a^{k'}\|. \tag{8}$$

Note that $L$-smoothness is defined with respect to the actions of other agents only, and not with respect to the central agent's own action $a^j$ or the state $s$.

**Theorem 1.** *When the multi-agent Q-function is additively decomposable, and it is L-smooth, then it can be approximated by the mean field Q-function, satisfying,*

$$|Q^j(s, \boldsymbol{a}) - Q^j(s, a^j, \bar{a}^j)| \leq L.$$

*Proof.* Let $Q^j(s, \boldsymbol{a})$ represent the multi-agent $Q$ function of the agent $j$. The multi-agent $Q$-function is additively decomposible which means that the $Q$-function can be factorized using only the pairwise interactions with other agents,

$$Q^j(s, \boldsymbol{a}) = \frac{1}{N} \sum_k Q^j(s, a^j, a^k), \tag{9}$$

where, $N$ denotes the number of other agents in the environment and $k$ is an agent index. Further, the condition of being $L$-smooth guarantees that the multi-agent $Q$-function has a bounded rate of change with

the Lipschitz constant of the gradient being $L$ (i.e., the difference between gradients at any two points is bounded by $L$ times the distance between those points).

As shown in Yang et al. (2018), the pairwise Q function $Q^j(s, a^j, a^k)$ if twice-differentiable w.r.t the action $a^k$ taken by another agent $k$, can be expanded as (with $N$ representing the number of agents in the environment and $\delta a^{j,k}$ being a small fluctuation of the action $a^k$ of agent $k$ from the mean field $\overline{a}^j$),

$$
\begin{aligned}
Q^j(s, \boldsymbol{a}) &= \tfrac{1}{N} \sum_k Q^j(s, a^j, a^k) \\
&= \tfrac{1}{N} \sum_k \left[ Q^j(s, a^j, \overline{a}^j) + \nabla_{\overline{a}^j} Q^j(s, a^j, \overline{a}^j) \cdot \delta a^{j,k} + \tfrac{1}{2} \delta a^{j,k} \cdot \nabla^2_{\tilde{a}^{j,k}} Q^j(s, a^j, \tilde{a}^{j,k}) \cdot \delta a^{j,k} \right] \\
&= Q^j(s, a^j, \overline{a}^j) + \nabla_{\overline{a}^j} Q^j(s, a^j, \overline{a}^j) \cdot \left[ \tfrac{1}{N} \sum_k \delta a^{j,k} \right] + \tfrac{1}{2N} \sum_k \left[ \delta a^{j,k} \cdot \nabla^2_{\tilde{a}^{j,k}} Q^j(s, a^j, \tilde{a}^{j,k}) \cdot \delta a^{j,k} \right] \\
&= Q^j(s, a^j, \overline{a}^j) + \tfrac{1}{2N} \sum_k R^j_{s, a^j}(a^k)
\end{aligned}
\tag{10}
$$

Here $R^j_{s, a^j}(a^k) \triangleq \delta a^{j,k} \cdot \nabla^2_{\tilde{a}^{j,k}} Q^j(s, a^j, \tilde{a}^{j,k}) \cdot \delta a^{j,k}$ denotes the Taylor Polynomial reminder. Also, $\tilde{a}^{j,k} = \overline{a}^j + \epsilon^{j,k} \delta a^{j,k}$, where $\epsilon^{j,k} \in [0,1]$. The term $\sum_k \delta a^{j,k} = 0$, which is applied in the last step.

Further, Yang et al. (2018) proved that the reminder term $R^j$ is upper bounded by $2L$, where $L$ is the Lipschitz constant.

Hence, from Eq. 10 we have that,

$$
|Q^j(s, \boldsymbol{a}) - Q^j(s, a^j, \overline{a}^j)| \leq L
\tag{11}
$$

Note, though Yang et al. (2018) consider a notion of neighbourhoods for similar arguments, we do not consider any notion of neighbourhoods. We will assume that all agents in the environment are part of the same "neighbourhood" considered by Yang et al. (2018).

$\square$

While Yang et al. (2018) argued that the approximation error is 0 when neighbourhoods are used, Theorem 1 demonstrates that there is indeed a finite error bound when neighbourhood assumptions are relaxed. From Theorem 1 we see that if it were tractable to use the multi-agent $Q$-function in the environment, then it should be used directly. If it is not tractable to use the multiagent $Q$-function then the MFQ can be used by trading off tractable computations for a small approximation error.

Next, we state some assumptions (the same as those in Yang et al. (2018)) and provide a convergence result in Theorem 2.

**Assumption 1.** *Every $s \in \mathcal{S}$ and $a^j \in A^j$, for each agent $j \in \{1, \ldots, N\}$, are visited infinitely often, and the reward function for each agent $j$ stays bounded.*

**Assumption 2.** *For all $s, t$, and $\boldsymbol{a}$, $0 \leq \alpha_t(s, \boldsymbol{a}) < 1$, $\sum_{t=0}^{\infty} \alpha_t(s, \boldsymbol{a}) = \infty$, $\sum_{t=0}^{\infty} [\alpha_t(s, \boldsymbol{a})]^2 < \infty$.*

**Assumption 3.** *For each agent $j$, the agent's policy is greedy in the limit with infinite exploration (GLIE).*

**Assumption 4.** *For each stage game at state $s$ and time $t$, NE is a global optimum or a saddle point.*

The first assumption is a standard condition that is needed for convergence of off-policy learning methods that guarantees that all states and actions are visited infinitely often (Szepesvári & Littman, 1999). The second assumption is the standard Robbins-Monro condition on the learning rate (Clark, 1984; Yang et al., 2018). The third assumption is the Greedy in the Limit with Infinite Exploration (GLIE) condition that mean field RL algorithms are guaranteed to satisfy (Yang et al., 2018). Finally the fourth assumption is a restriction on the nature of stochastic games that guarantees that these classes of stochastic games have a unique equilibrium point. This guarantees a unique solution concept which makes it possible to apply contraction based results commonly found in the single-agent RL literature (Szepesvári & Littman, 1999).

**Theorem 2.** *Given a parameterized joint policy $\boldsymbol{\pi}_{\phi_t}$, with a set of features $\{\omega_p, p = 1, \ldots, P\}$, let the joint policy be $\frac{\mathcal{K}}{2T}$ Lipschitz continuous with respect to $\phi_t$. Given Assumptions 1– 4, there exists a constant $C_0$ such that both MFA-QL and MFAA-QL converge to NE w.p. 1 if $\frac{\mathcal{K}}{2T} < C_0$.*

*Proof.* The proof of this theorem for MFA-QL and MFAA-QL follows the same steps as in Theorem 2 of Yang et al. (2018), once all the conditions are satisfied. Based on the exploration conducted by MFA-QL and MFAA-QL, the first assumption is satisfied. The learning rate conditions (Assumption 2) is satisfied based on the learning rate being set to be inverse of the number of time steps $t$ in MFA-QL and MFAA-QL. The Assumption 3 is satisfied due to the exploration rate being decayed to 0 at the end of training. Further, for MFAA-QL, the action advising follows the probabilistic policy reuse strategy (Fernández & Veloso, 2006) which ensures that the action advising also decays to 0 at the end of training. Assumption 4 is a restriction on the type of stage game, which we assume is satisfied by the nature of the environment. As specified by Yang et al. (2018), satisfying this assumption is extremely hard in practice, however it is required to prove convergence in theory. Further, according to Hu & Wellman (2003) and Yang et al. (2018), convergence is still observed when this assumption is violated in practice. □

## H  Mean Field Attention - Actor Critic

In this section we provide the complete pseudocode of the Mean Field Attention Actor Critic algorithm (MFA-AC). This extends MFA-QL to the actor-critic method where the $Q$-function serves as an critic and the policy derived from $Q$ as the actor. The pseudocode is in Algorithm 2.

## I  Integrating Action Advising in MFA

In this section, we provide the algorithmic extension of MFA to the mean field attention advising (MFAA) algorithm, which accelerates training in MFA using pre-existing sources of knowledge. These pre-existing sources of knowledge can be pre-trained networks, physics-based policies, or control policies (broadly denoted as *advisors* in Subramanian et al. (2022)).

The action advising component of MFAA follows the same scheme as in Subramanian et al. (2022), where the probabilistic policy reuse (PPR) (Fernández & Veloso, 2006) technique is used to incorporate a finite amount of action advice from an online advisor during training. As in Subramanian et al. (2022), we assume that every agent can have access to at-most one advisor. The PPR technique uses a hyperparameter ($\epsilon'$) to control the probability of receiving action advice from an advisor. This hyperparameter is decayed (linearly) in such a way that the agent receives maximum advisor inputs at the beginning of training and subsequently, the frequency of advisor inputs is gradually reduced. Whenever the action advice from the advisor is available, the agent performs this action as opposed to relying on its own policy. After a fixed number of training episodes, the agent will not receive any further inputs from the advisor and must rely on its own policy for action selection. The intuition here is that an agent gets maximum help from the advisor during earlier stages of learning, when its own policy is untrained and close to random. Later on in training, when its own policy improves, the agent is expected to rely more on its own policy for actions and less on the advisor. At the end of training, the agent should learn to act fully independent of the advisor. The agent is expected to be deployed independently without any advisor.

The rest of the algorithmic steps of MFAA follows the same scheme as in MFA. The complete pseudocode is provided in Algorithm 3.

## J  Extension of MFAA to Multi-Type Mean Field Reinforcement Learning

### J.1  Background

The homogeneity assumptions in MFRL are strong and are not very applicable in real-world environments as these environments usually contain diverse sets of agents that are different in objectives and abilities. In this context, Subramanian et al. (2020) extend the MFRL framework to environments that may contain

---

**Algorithm 2** Mean Field Attention Actor-Critic (MFA-AC)

---

1: Initialize the $Q$ functions (parameterized by weights) $Q_{\phi^j}, Q_{\phi^j_{\_}}$, and the policies $\pi_{\theta^j}, \pi_{\theta^j_{\_}}$, for all agents $j$ configured according to Eq. 6
2: Initialize the mean action $\overline{a}^j$ for each agent $j \in \{1, \ldots, N\}$ and the size of importance set $K^j$
3: Initialize the total number of steps (T), the total number of episodes (E), and initial state $s$
4: **while** Episode < E **do**
5:    **while** Step < T **do**
6:       **while** $j = 1$ to $N$ **do**
7:          Choose action $a^j$ from $\pi_{\theta^j}(s)$ according to Eq. 4 using the exploration rate $\beta$
8:          Compute the new mean action $\overline{a}^j$ according to Eq. 5 for agents outside the importance set
9:       **end while**
10:       Execute the joint action $\boldsymbol{a} = [a^1, \ldots, a^N]$. Observe the rewards $\boldsymbol{r} = [r^1, \ldots, r^N]$ and the next state $s'$
11:       Store $\langle s, \boldsymbol{a}, \boldsymbol{r}, \boldsymbol{k}, s', \overline{\boldsymbol{a}} \rangle$ in replay buffer $D$, where $\overline{\boldsymbol{a}} = [\overline{a}^1, \ldots, \overline{a}^N]$ is the joint mean action, and $\boldsymbol{k} = [k^1, \ldots, k^N], \mathcal{I}^j = [o^1, a^1, \ldots, o^{K^j}, a^{K^j}]$ denotes the importance set of $j$
12:       Assign $s \leftarrow s'$
13:    **end while**
14:    **while** $j = 1$ to $N$ **do**
15:       Sample a minibatch of $M$ experiences $\langle s, \boldsymbol{a}, \boldsymbol{r}, \boldsymbol{k}, s', \overline{\boldsymbol{a}} \rangle$ from $D$ and store it in $\mathcal{D}$
16:       Using $\mathcal{D}$, set $y^j = r^j + \gamma v^{MF}_{\phi^j_{\_}}(s')$ according to Eq. 1
17:       Using $\mathcal{D}$, update the $Q$ network (critic) by minimizing the loss $L(\phi^j) = \frac{1}{M} \sum (y^j - Q_{\phi^j}(s, a^j, \overline{a}^j))^2$
18:       Using $\mathcal{D}$, update the actor using the sample policy gradient:

$$\nabla_{\theta^j} J(\theta^j) \approx \frac{1}{K} \sum \nabla_{\theta^j} \log \pi_{\theta^j}(s'^j) Q_{\phi^j_{\_}}(s', a^j_{\_}, \overline{a}^j)|_{a^j_{\_} = \pi_{\theta^j_{\_}}(s'^j)} \tag{12}$$

19:    **end while**
20:    Update the parameters of the target network for each agent $j$ with learning rate $\tau_\phi$ and $\tau_\theta$;

$$\phi^j_{\_} \leftarrow \tau_\phi \phi^j + (1 - \tau_\phi) \phi^j_{\_}$$

$$\theta^j_{\_} = \tau_\theta \theta^j + (1 - \tau_\theta) \theta^j_{\_}$$

21:    Linearly decay the value of $\beta$
22: **end while**

---

heterogeneous agents. The approach groups agents into a finite set of *types*, where these types are constructed in such a way that the mean field assumption of homogeneity is valid within types but not applicable across types. Assuming the presence of $M$ agent types, the mean field update equations in Eqs. $1 - 4$ are replaced with the multi-type mean field (MTMF) updates, given as follows:

$$Q^j(s_t, a^j_t, \overline{a}^j_{1,t}, \ldots, \overline{a}^j_{M,t}) = (1 - \alpha) Q^j(s_t, a^j_t, \overline{a}^j_{1,t}, \ldots, \overline{a}^j_{M,t}) + \alpha[r^j_t + \gamma v^j(s_{t+1})] \tag{13}$$

where,

$$v^j(s_{t+1}) = \sum_{a^j_{t+1}} \pi^j(a^j_{t+1} | s_{t+1}, \overline{a}^j_{1,t}, \ldots, \overline{a}^j_{M,t}) Q^j(s_{t+1}, a^j_{t+1}, \overline{a}^j_{1,t}, \ldots, \overline{a}^j_{M,t}) \tag{14}$$

$$\overline{a}^j_{i,t} = \frac{1}{N_i} \sum_{k \neq j} a^k_{i,t}, \quad a^k_{i,t} \sim \pi^k(\cdot | s_t, \overline{a}^k_{1,t-1}, \ldots, \overline{a}^k_{M,t-1}) \tag{15}$$

$$\pi^j(a^j_t | s_t, \overline{a}^j_{1,t-1}, \ldots, \overline{a}^j_{M,t-1}) = \frac{\exp(-\beta Q^j(s_t, a^j_t, \overline{a}^j_{1,t-1}, \ldots, \overline{a}^j_{M,t-1}))}{\sum_{a^{j'}_t \in A^j} \exp(-\beta Q^j(s_t, a^{j'}_t, \overline{a}^j_{1,t-1}, \ldots, \overline{a}^j_{M,t-1}))} \tag{16}$$

---

**Algorithm 3** Mean Field Attention Advising $Q$-Learning (MFAA-QL)

---

1: Initialize the $Q$ functions (parameterized by weights) $Q_{\phi^j}, Q_{\phi^j\_}$, for all agents $j$ configured according to Eq. 6
2: Initialize the mean action $\bar{a}^j$ for each agent $j \in \{1, \dots, N\}$ and the size of importance set $K^j$
3: Initialize a value for the hyperparameter $\epsilon'$ (i.e. value for $\epsilon'_0$)
4: Initialize the total number of steps (T), total number of episodes (E), and initial state $s$
5: **while** Episode $<$ E **do**
6:   **while** Step $<$ T **do**
7:     **while** $j = 1$ to $N$ **do**
8:       Let $u$ be a uniform random number between 0 and 1
9:       **if** $u < \epsilon'_t$ **then**
10:         Obtain action $a^j$ from the advisor (using current state $s$ and mean action $\bar{a}^j$)
11:       **else**
12:         Choose action $a^j$ from $Q_{\phi^j}$ according to Eq. 4 using the mean action $\bar{a}^j$, state $s$, and the exploration rate $\beta$
13:       **end if**
14:       Compute the new mean action $\bar{a}^j$ according to Eq. 5 for agents outside the importance set for agents outside the importance set of $j$
15:     **end while**
16:     Execute the joint action $\boldsymbol{a} = [a^1, \dots, a^N]$. Observe the rewards $\boldsymbol{r} = [r^1, \dots, r^N]$ and the next state $s'$
17:     Store $\langle s, \boldsymbol{a}, \boldsymbol{r}, \boldsymbol{k}, s', \overline{\boldsymbol{a}} \rangle$ in replay buffer $D$, where $\overline{\boldsymbol{a}} = [\bar{a}^1, \dots, \bar{a}^N]$ is the joint mean action, and $\mathcal{I}^j = [k^1, \dots, k^N]$, $k^j = [o^1, a^1, \dots, o^{K^j}, a^{K^j}]$ denotes the importance set of $j$
18:     Assign $s \leftarrow s'$
19:   **end while**
20:   **while** $j = 1$ to $N$ **do**
21:     Sample a minibatch of $M$ experiences $\langle s, \boldsymbol{a}, \boldsymbol{r}, \boldsymbol{k}, s', \overline{\boldsymbol{a}} \rangle$ from $D$ and store it in $\mathcal{D}$
22:     Using $\mathcal{D}$, set $y^j = r^j + \gamma v_{\phi^j\_}^{MF}(s')$ according to Eq. 1
23:     Using $\mathcal{D}$, update the $Q$ network by minimizing the loss $L(\phi^j) = \frac{1}{M} \sum (y^j - Q_{\phi^j}(s, a^j, \bar{a}^j))^2$
24:   **end while**
25:   Update the parameters of the target network for each agent $j$ with learning rate $\tau$;

$$\phi^j\_ \leftarrow \tau \phi^j + (1 - \tau) \phi^j\_$$

26:   At the end of each episode linearly decay $\epsilon'_t$
27:   Linearly decay the value of $\beta$
28: **end while**

---

Here, the notation $\bar{a}^j_{i,t}$ denotes the mean field of agent $j$ belonging to type $i$ at time $t$. The variable $N_i$ denotes the number of agents belonging to type $i$. All the other variables have the same meaning as in Eqs. 1 – 4. Subramanian et al. (2020) provide theoretical convergence guarantees and bounds on approximation errors in mean field environments with multiple types, while using these MTMF update equations. Furthermore, Subramanian et al. (2020) provide a practical algorithm, multi-type mean field $Q$-learning (MTMFQ), that uses function approximation (similar to the approach in MFQ), and show that this algorithm out-performs MFQ in different many-agent heterogeneous environments.

## J.2 Multi-Type Mean Field Attention Advising

In this sub-section, we extend the MFA framework to include multiple types using the approach in Subramanian et al. (2020). Assuming a total of $M$ types in the environment (indexed by $m$, i.e. $m \in [1, \dots, M]$), we use the update equations provided in Eqs. 13 – 16. While Subramanian et al. (2020) analyze two different kinds of type-based mean field environments, i.e. known types and unknown types, we restrict our work to

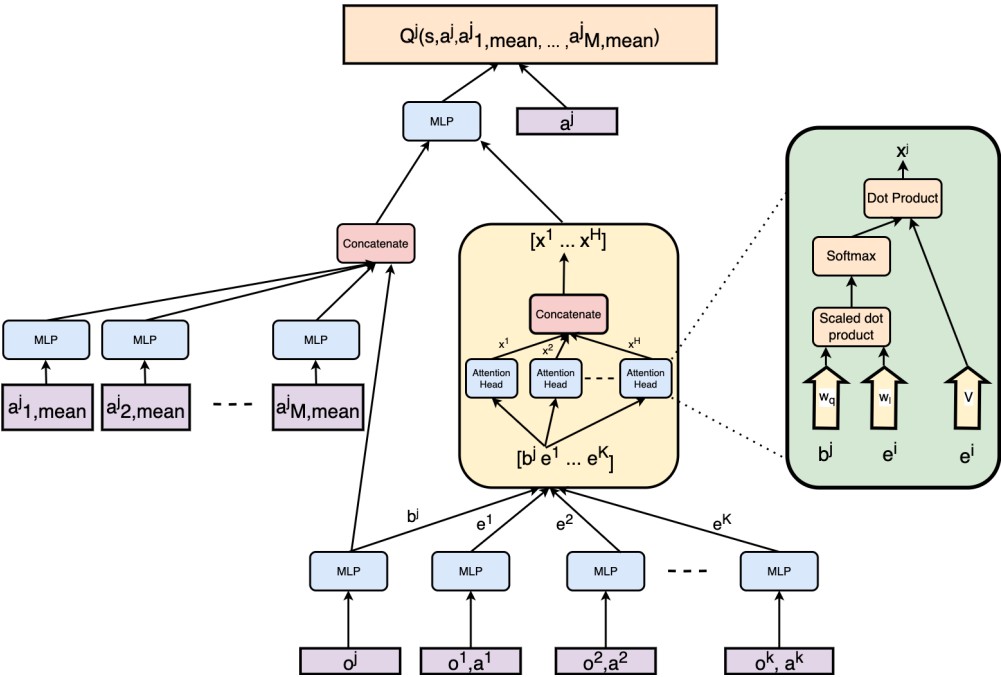

Figure 17: Multi-type network architecture for calculating (function approximation based value) $Q^j(s, a^j, \overline{a}^j_{1,t}, \ldots, \overline{a}^j_{M,t})$ for each agent $j$. As compared to Figure 1, this version of our algorithm considers different types, so the mean field of each type is explicitly handled (as opposed to just a single mean field in Figure 1).

the case of known types where the type classification of an agent in the environment is known prior to the commencement of the game (public knowledge). The extension to unknown types is straight forward, and we omit the associated discussions.

The only change we need to make in our MFA algorithm is to include the mean field action of each type separately in the neural network architecture provided in Figure 1. The updated architecture is provided in Figure 17, where the mean fields of the different types are maintained separately.

Now, we provide a practical $Q$-learning algorithm for the multiple type-based updates called Multi-Type Mean Field Attention $Q$-Learning (MTMFA-QL). The complete pseudocode for MTMFA-QL is provided in Algorithm 4. We also extend this method to an actor-critic technique in Algorithm 5, denoted as Multi-Type Mean Field Attention Actor-Critic (MTMFA-AC).

Using the same procedure as that of the extension of MFA to MFAA using a probabilistic policy reuse (PPR) based action advising technique (Subramanian et al., 2022), MTMFA-QL can be extended to Multi Type Mean Field Attention Advising $Q$-learning (MTMFAA-QL), which accelerates training using action advising from a pre-existing external source of knowledge (broadly *advisors*). This technique also integrates the PPR based technique from Subramanian et al. (2022) exactly in the same way as MFAA.

---

**Algorithm 4** Multi-Type Mean Field Attention $Q$-Learning (MTMFA-QL)

---

1: Initialize the $Q$ functions (parameterized by weights) $Q_{\phi^j}, Q_{\phi^j_-}$, for all agents $j$ configured according to Eq. 6
2: Let the environment have $M$ types Initialize the mean action for each type $\overline{a}^j_1, \ldots, \overline{a}^j_M$ for each agent $j \in \{1, \ldots, N\}$ and the size of importance set $K^j$
3: Initialize the total number of steps (T), the total number of episodes (E), and initial state $s$
4: **while** Episode $<$ E **do**
5:   **while** Step $<$ T **do**
6:     **while** $j = 1$ to $N$ **do**
7:       Choose action $a^j$ from $Q_{\phi^j}$ according to Eq. 16 using the mean action for each type $\overline{a}^j_1, \ldots, \overline{a}^j_M$, state $s$, and the exploration rate $\beta$
8:       For each agent j, compute the new mean action for each type $\overline{a}^j_1, \ldots, \overline{a}^j_M$ according to Eq. 15 for agents outside the importance set of $j$
9:     **end while**
10:   Execute the joint action $\boldsymbol{a} = [a^1, \ldots, a^N]$. Observe the rewards $\boldsymbol{r} = [r^1, \ldots, r^N]$ and the next state $s'$
11:   Store $\langle s, \boldsymbol{a}, \boldsymbol{r}, \boldsymbol{k}, s', \overline{\boldsymbol{a}}_1, \ldots, \overline{\boldsymbol{a}}_M \rangle$ in replay buffer $D$, where $\overline{\boldsymbol{a}}_i = [\overline{a}^1_i, \ldots, \overline{a}^N_i]$ is the joint mean action for type $i$, and $\boldsymbol{k} = [k^1, \ldots, k^N]$, $\mathcal{I}^j = [o^1, a^1, \ldots, o^{K^j}, a^{K^j}]$ denotes the importance set of $j$. The $\boldsymbol{a}$ captures all the $N$ agents
12:   Assign $s \leftarrow s'$
13:   **end while**
14:   **while** $j = 1$ to $N$ **do**
15:     Sample a minibatch of $M$ experiences $\langle s, \boldsymbol{a}, \boldsymbol{r}, \boldsymbol{k}, s', \overline{\boldsymbol{a}}_1, \ldots, \overline{\boldsymbol{a}}_M \rangle$ from $D$ and store it in $\mathcal{D}$
16:     Using $\mathcal{D}$, set $y^j = r^j + \gamma v^{MTMF}_{\phi^j_-}(s')$ according to Eq. 13
17:     Using $\mathcal{D}$, update the $Q$ network by minimizing the loss

$$L(\phi^j) = \frac{1}{M} \sum (y^j - Q_{\phi^j}(s, a^j, \overline{a}^j_1, \ldots, \overline{a}^j_M))^2$$

18:   **end while**
19:   Update the parameters of the target network for each agent $j$ with learning rate $\tau$;

$$\phi^j_- \leftarrow \tau \phi^j + (1 - \tau)\phi^j_-$$

20:   At the end of each episode linearly decay the value of $\beta$
21: **end while**

---

---

**Algorithm 5** Multi-Type Mean Field Attention Advising Actor-Critic (MTMFA-AC)

---

1: Initialize the $Q$ functions (parameterized by weights) $Q_{\phi^j}, Q_{\phi^j_{\_}}$, and the policies $\pi_{\theta^j}, \pi_{\theta^j_{\_}}$, for all agents $j$, configured according to Eq. 6

2: Let the environment have $M$ types. Initialize the mean action for each type $\bar{a}^j_1, \ldots, \bar{a}^j_M$ for each agent $j \in 1, \ldots, N$, and the size of importance set $K^j$

3: Initialize the total number of steps (T), total number of episodes (E), and initial state $s$

4: **while** Episode $<$ E **do**

5:    **while** Step $<$ T **do**

6:       **while** $j = 1$ to $N$ **do**

7:          Choose action $a^j$ from $\pi_{\theta^j}(s)$ according to Eq. 4 using the exploration rate $\beta$

8:          For each agent j, compute the new mean action for each type $\bar{a}^j_1, \ldots, \bar{a}^j_M$ for agents outside the importance set of $j$, according to Eq. 15

9:       **end while**

10:       Execute the joint action $\boldsymbol{a} = [a^1, \ldots, a^N]$. Observe the rewards $\boldsymbol{r} = [r^1, \ldots, r^N]$ and the next state $s'$

11:       Store $\langle s, \boldsymbol{a}, \boldsymbol{r}, \boldsymbol{k}, s', \bar{\boldsymbol{a}}_1, \ldots, \bar{\boldsymbol{a}}_M \rangle$ in replay buffer $D$, where $\bar{\boldsymbol{a}}_i = [\bar{a}^1_i, \ldots, \bar{a}^N_i]$ is the joint mean action for type $i$, and $\boldsymbol{k} = [k^1, \ldots, k^N]$, $\mathcal{I}^j = [o^1, a^1, \ldots, o^{K^j}, a^{K^j}]$ denotes the importance set of $j$. The $\boldsymbol{a}$ captures all the $N$ agents

12:       Assign $s \leftarrow s'$

13:    **end while**

14:    **while** $j = 1$ to $N$ **do**

15:       Sample a minibatch of $M$ experiences $\langle s, \boldsymbol{a}, \boldsymbol{r}, \boldsymbol{k}, s', \bar{\boldsymbol{a}}_1, \ldots, \bar{\boldsymbol{a}}_M \rangle$ from $D$ and store it in $\mathcal{D}$

16:       Using $\mathcal{D}$, set $y^j = r^j + \gamma v^{MTMF}_{\phi^j_{\_}}(s')$ according to Eq. 13

17:       Using $\mathcal{D}$, update the $Q$ network (critic) by minimizing the loss $L(\phi^j) = \frac{1}{M} \sum (y^j - Q_{\phi^j}(s, a^j, \bar{a}^j_1, \ldots, \bar{a}^j_M))^2$

18:       Update the actor using the sample policy gradient:

$$\nabla_{\theta^j} J(\theta^j) \approx \frac{1}{K} \sum \nabla_{\theta^j} \log \pi_{\theta^j}(s'^j) Q_{\phi^j_{\_}}(s', a^j_{\_}, \bar{a}^j_1, \ldots, \bar{a}^j_M)|_{a^j_{\_}=\pi_{\theta^j_{\_}}(s'^j)}$$

19:    **end while**

20:    Update the parameters of the target network for each agent $j$ with learning rate $\tau_\phi$ and $\tau_\theta$;

$$\phi^j_{\_} \leftarrow \tau_\phi \phi^j + (1 - \tau_\phi)\phi^j_{\_}$$

$$\theta^j_{\_} = \tau_\theta \theta^j + (1 - \tau_\theta)\theta^j_{\_}$$

21:    At the end of each episode linearly decay the value of $\beta$

22: **end while**

---

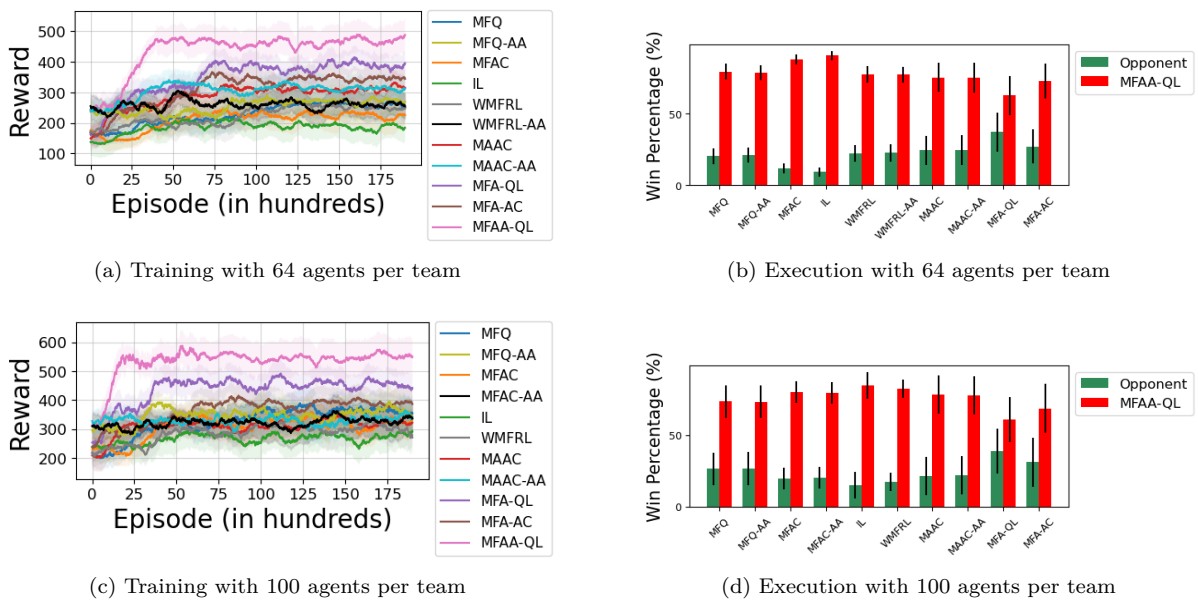

(a) Training with 64 agents per team        (b) Execution with 64 agents per team

(c) Training with 100 agents per team        (d) Execution with 100 agents per team

Figure 18: MAgent Battle environment results for Action Advising.

## K    Additional Results on Action Advising

While the results in Section 5 considered action advising for the MFA algorithms, here we provide additional results incorporating action advising for mean field methods from prior works that we used as baselines. Specifically, we consider the 64-agent and 100-agent versions of the MAgent Battle game we used in Section 5 and we incorporate action advising to the three best performing baseline algorithms from prior works in each domain. The advising comes from a pre-trained IL network (trained for 100000 episodes), as done in Section 5. We use the same approach as described in Appendix I, where we use the PPR technique to incorporate a finite amount of action advice from an online advisor.

Training and execution experiments are performed as in Section 5. For the 64-agent version, we incorporate action advising with algorithms MFQ, WMFRL, and MAAC (algorithms denoted as MFQ-AA, WMFRL-AA, and MAAC-AA, respectively). We select these three algorithms as they were the three best performing baselines from prior works in our experiments in Section 5. Similarly, for the 100-agent version, we incorporate action advising with algorithms MFQ, MFAC, and MAAC (algorithms denoted as MFQ-AA, MFAC-AA, and MAAC-AA, respectively). The training and execution results are in Figure 18. From the results, we note that the action advising does provide a warm start for the algorithms as expected, but does not lead to any significant gains in performance towards the end of training as compared to the implementation that does not use any action advising. For instance, in Figure 18(a) we note that MAAC and MAAC-AA largely converge to providing very similar performance at the end of training and similarly, MFQ and MFQ-AA converge to providing very similar performance at the end of training. Even during execution, these methods provide a very similar performance. The reason for this observation is that these prior methods have some fundamental limitations in large agent environments (as explained in Section 5, that leads to converging towards a comparitively poor local optimum), which cannot be overcome by action advising. To recall, purely mean field methods that do not use attention (such as MFQ and MFAC) cannot give higher importance to nearby agents based on need, and that leads to a poor performance. Methods that ignore agents further away (such as WMFRL) misses crucial context for decision making, and methods that only use the attention mechanism without the mean field (such as MAAC) includes unnecessary detail that makes it difficult to focus on the right information for decision making and leads to other problems (like their inability to scale to environments with more agents).

