# OpenReview forum: "Revisiting Neighbourhoods in Mean Field Reinforcement Learning"
_TMLR — Decision pending for TMLR_

### Review · Reviewer_DuYF · 2026-04-12

**Summary Of Contributions:**

This paper proposes a novel mean-field MARL method called mean-field attention (MFA) which interpolates Nash Q function and mean-field Q function to tradeoff speed and approximation error. The main idea is to split the naive mean-field assumption into an importance set constructed based on a notion of distance and the rest remain as the mean-field set so that the importance set better approximates the Nash Q function. An attention architecture is used to represent the importance set. The authors also introduced a variant of MFA with action advising. Experiment results on a variety of environments show strong performance against a variety of baselines.

**Audience:**

Yes

**Audience Explanation:**

This paper would be of interest to the TMLR audience because it address an important scaling problem in MARL with a sensible method. The empirical and practical advantage of the proposed method is clear in terms of both performance and wall-clock time.

**Claims And Evidence:**

No

**Claims Explanation:**

The main claim of the paper is that MFA better approximates the Nash Q function by relaxing the naive mean-field structure while maintaining tractability. The paper demonstrated strong empirical results, but I believe there are some missing pieces which make the claim not fully supported. I am willing to update this assessment after the authors address these concerns.

* The proposed MFA variants clearly have better training performance in terms of reward, but their win percentages are often not high. Could the authors explain this gap?
* To show that MFA is a better approximation than naive mean-field, I believe it is useful to present a simple setting where the ground truth Nash Q can be computed (approximately), such as in the Ising model. In that case the authors can not only evaluate the approximation error but also visualize the approximated value function.
* The connection between the proposed method and the theoretical results is not clear. I will detail more on this point in the requested changes section.

**Requested Changes:**

* Theoretical results gap: it is not clear to me how the theoretical results in section 5, especially theorem 1, is fundamentally different from what's already presented in Yang et al, 2018. You essentially use the same proof except that Yang et al considered the second order term as negligible. It's not clear to me that whether you keeping this term is motivated by the MFA structure. If not then I am unsure of the utility of this theorem.
* As said above, it would be good to explain why higher training reward does not imply higher win rate. In that case there's potentially a gap in the training vs eval criteria?
* The authors discussed heterogeneous agents as an important feature in the proposed method and presented experiment 3. However, the experiment setup does not directly show that MFA has a particular advantage in heterogeneous agent environments or that other methods particularly suffer in heterogeneous agent environments, since the results are not presented with a same environment no heterogeneous baseline.
* The effect of action advising can be particularly strong in many environments. However, it seems to me that action advising can also be applied to other baseline algorithms, in which case the performance ranking many be altered. Could the authors comment on this and why AA is only applied to MFA?

---

> ### Author Response · Authors · 2026-04-22
> **Response to the Reviewer 1/2**
>
> **Performances - Rewards vs Win Rate:**
>
> We thank the reviewer for this question. The apparent gap between training rewards and win percentages arises because training and execution are designed to test different characteristics of the algorithms, and there is no expectation that higher training rewards will directly translate to higher win percentages.
>
> During training, all algorithms are trained using self-play, with no knowledge of which algorithm they will face during execution. This is a deliberate design choice that tests the generalizability of the learned policies. During execution, the trained policies are evaluated in a face-off context against MFAA-QL, the best performing algorithm during training, with no further training, exploration, or action advising. The execution phase therefore tests how well a policy learned in self-play generalizes to facing a specific strong opponent — a different characteristic from training performance. This experimental methodology is consistent with the most closely related prior works in the MFRL literature, including Yang et al. (2018), Subramanian et al. (2020), Wang et al. (2022), and Ma et al. (2023).
>
> Importantly, we note that the general trends observed during training are indeed reflected in the execution experiments, which validates the consistency of our results. For example, in the 64-agent Battle game, the training performance ranking of MFAA-QL > MFA-QL > MFA-AC > MAAC > MFQ $\approx$ WMFRL > MFAC > IL is reflected in the execution win percentages as well. Similarly, in the 100-agent Battle game, the training ranking of MFAA-QL > MFA-QL > MFA-AC > MFQ > MFAC $\approx$ MAAC > WMFRL $\approx$ IL is also consistent with the execution results. This demonstrates that while the absolute win percentages may not be high for all algorithms (which is expected given that they are facing the strongest algorithm) the relative ordering of algorithms is preserved across both phases, providing a consistent and coherent picture of the relative merits of each approach.
>
> **Theoretical Results:**
>
> While Yang et al. (2018) come very close to providing a theorem, they do not provide a formal result in their paper that explicitly states the conditions required for an approximation bound while using the mean field approximation. They simply state that the approximation error is 0 when the locality and homogeneity assumptions are used, without formalizing this as a theorem with explicit assumptions. In our work, we provide a formal result that explicitly states the necessary assumptions and derives the approximation bound when the locality and homogeneity assumptions are relaxed, which is the setting MFA operates in. That said, we acknowledge that this difference is relatively modest in terms of proof novelty, and this is precisely why we have moved the theoretical results to the appendix. The revised draft better reflects the primarily empirical nature of our contributions.
>
>  **Heterogeneous Agent Experiments:**
>
> We thank the reviewer for this important observation. We note that the core message that homogeneous algorithms do not perform well in heterogeneous agent environments was already established by Subramanian et al. (2020), who demonstrated this across a variety of heterogeneous many-agent environments. This is why these comparisons were not included in the original submission. Nonetheless, we agree with the reviewer that including these comparisons directly in the same experimental setting strengthens the paper, and we have added them in the revised draft.
>
> Specifically, we have added homogeneous baselines (MFQ, MFAC, and MFA-QL) to Experiment 3 (Combined Arms heterogeneous environment) in the revised paper. These algorithms assume fully homogeneous agents and do not incorporate type classification, allowing for a direct comparison between methods that account for heterogeneity and those that do not within the same environment. The results in the revised Figure 5 clearly confirm that algorithms that do not consider types and assume homogeneous agents (MFQ, MFAC, and MFA-QL) perform very poorly in this experiment, consistent with the observations in Subramanian et al. (2020). Notably, while MFA-QL outperforms MFQ in this experiment, its performance is inferior to MTMFQ, which considers types but does not use the attention mechanism. This shows that for heterogeneous agent environments, incorporating types is more important than using the attention mechanism alone, serving as a strong motivation for the type-based MFA algorithms (MTMFA-QL, MTMFA-AC, and MTMFAA-QL) provided in this work.

---

> > ### Author Response · Authors · 2026-04-22
> > **Response to the Reviewer 2/2**
> >
> > **Action Advising to baseline algorithms:**
> >
> > We thank the reviewer for this important question. We note that action advising was not applied to baseline algorithms in the original manuscript because our preliminary experiments showed that action advising does not provide significant gains for the baseline algorithms from prior works beyond a warm start effect. Nonetheless, we agree with the reviewer that including these results directly in the paper strengthens our claims, and we have added them in the revised draft in Appendix K.
> >
> > Specifically, in Appendix K we present results incorporating action advising for the three best performing baseline algorithms in the 64-agent and 100-agent Battle environments (MFQ-AA, WMFRL-AA, and MAAC-AA for 64 agents, and MFQ-AA, MFAC-AA, and MAAC-AA for 100 agents). The results in Figure 18 show that while action advising does provide a warm start for these algorithms as expected, it does not lead to any significant gains in performance towards the end of training compared to their implementations without action advising. For instance, MAAC and MAAC-AA largely converge to very similar performance at the end of training, and the same is observed for MFQ and MFQ-AA. The performance ranking across all algorithms is also preserved in the execution experiments. This demonstrates that the performance advantages of MFAA-QL over baseline algorithms are due to the fundamental design of MFA, i.e., combining attention for local responses, mean field approximation for global responses, and not ignoring any agents, rather than being an artifact of action advising alone. The limitations of the baseline algorithm from prior works are fundamental in nature, and cannot be overcome by action advising, which confirms that the performance ranking observed in the original manuscript remains unchanged.
> >
> >
> >
> > **Ising Model:**
> >
> > We thank the reviewer for this suggestion. However, we respectfully argue that the Ising model is not the appropriate setting for evaluating the advantages of MFA over a naive mean field algorithm. In the Ising model, as used in Figure 1 of Yang et al. (2018), the spin/action of each agent depends only on a finite number of fixed neighbours, and agents outside this neighbourhood have no influence on the central agent. This is precisely the setting where the naive MFRL formulation of Yang et al. (2018) is designed to work well, i.e., the locality and homogeneity assumptions are satisfied by construction, and the mean field approximation within the neighbourhood is sufficient. MFA is specifically designed for settings where these assumptions do not hold, i.e., where agents move freely, interaction structure changes dynamically, and distant agents can have non-negligible influence. Evaluating MFA in the Ising model would therefore not reveal its advantages over a naive mean field algorithm, and would in fact be expected to show little difference between the two approaches.
> >
> > We would also like to clarify a subtle but important point regarding the reviewer's characterization of our claims. The paper does not claim that MFA provides a better approximation than the naive mean field algorithm in terms of approximation error. Rather, the key advantage of MFA is that it provides a richer and more informative input representation to the Q-function compared to naive mean field, by explicitly modelling the differential influence of nearby agents through the attention mechanism while still accounting for all distant agents through the mean field. Naive mean field treats all agents uniformly through a single averaged signal, which loses important local information about nearby agents that is critical for effective decision making in many environments. The empirical results across five domains and three testbeds demonstrate that this richer input representation leads to consistently better performance than naive mean field approaches, which is the primary claim of the paper.

---

> > > ### Comment · Reviewer_DuYF · 2026-04-27
> > >
> > > Thank the authors for their responses. I appreciate the additional ablations and most of my questions are answered.
> > >
> > > * Regarding reward and win rate, I now realize you let each algorithm complete with the best algorithm. If you don't yet have a sentence on how to read plots like Figure 2 (b), maybe you can add one to make it easier to read.
> > > * Regarding the Ising model comment, my intention was not to use exactly the Ising model but a small scale diagnostic setting where the advantage of the proposed MFA approximation can be visualized or intuitively understood, for example by comparing with a computable oracle.

---

> > > > ### Author Response · Authors · 2026-04-27
> > > > **Reply to Reviewer DuYF**
> > > >
> > > > We thank the reviewer for these follow-up comments. We are happy to note that our additional experiments and responses in the revised draft have helped answer the prior questions of the reviewer. Furthermore, we would like to thank the reviewer once again for their careful review and raising these points which gave us an opportunity to improve the paper. We address the two remaining points below:
> > > >
> > > > Regarding the reward vs win rate plots, the details are provided under our description and explanation of Experiment 1 in Section 5. We will add another clarifying sentence for the camera-ready version to reinforce that each algorithm is evaluated in a face-off against MFAA-QL, the best performing algorithm during training, to make Figure 2(b) and similar execution plots easier to read.
> > > >
> > > > Regarding the suggestion of a small scale diagnostic setting, we would like to emphasize again that the key advantage of MFA over naive mean field is that it provides a richer and more informative input representation to the Q-function by explicitly modelling the differential influence of nearby agents through the attention mechanism while still accounting for all distant agents through the mean field. This advantage is fundamentally tied to large scale settings with many agents where neighbourhood effects are important and the interaction structure changes dynamically. In such settings, naive mean field loses important local information about nearby agents by treating all agents uniformly through a single averaged signal. In small scale diagnostic environments where the Nash Q-function can be computed exactly, the number of agents is necessarily small, which means the mean field approximation itself provides little benefit over exact methods, and the differential influence of nearby vs distant agents that MFA is designed to capture would not be apparent. This is precisely why MFA was designed and evaluated in large scale environments. Furthermore, we have provided a detailed analysis in Figure 3 using the Battle game, where we break down agent behaviours such as attack rates and kill rates, to provide the intuition that helps understand the advantages of MFA.

---

### Review · Reviewer_VDMA · 2026-04-15

**Summary Of Contributions:**

## Summary

This paper considers the problem of multi-agent reinforcement learning,
with an eye on designing methods that scale gracefully as the number of
agents increases. The contribution is placed on an interesting point in
the landscape between no zero compression of other agent behavior in the
Q-function – which is computationally expensive – vs high compression
(constant-space) of the other agent behavior via mixing / mean field
from prior works. In this work, it is proposed to consider fixed-sized
dynamic *importance sets* of agents relative to any given agent,
determined by some predefined notion of nearness between agents; then, a
mean field is constructed via an attention mechanism applied to the
states/actions of the agents in the importance set. This
(computationally) scales gracefully with the number of agents, and also
adaptively weighs the influence of agents finely in the construction of
the mean field. Through experiments in several domains, the proposed
approach outperforms a slew of baselines.

## Strengths

The proposed idea is fairly simple and a natural intervention on
existing approaches in the field.

The experimental results look quite strong, with the proposed approach
performing very well across several environments and against several
baselines with respect ot win rate, which is impressive.

## Weaknesses

There are a few weaknesses worth pointing out. Namely:

1.  The theoretical results are very weak, to the point where I believe
    they are unnecessary (see "requested changes"). Realistically, this
    proposed method is largely a heuristic. I think that's fine for now,
    but at least from my perspective, the theorems do not add anything
    to the paper.
2.  There are some assumptions that can be quite restrictive which are
    glossed over (again, see "requested changes"). Particularly, the
    assumption that there is a known distance function between agents is
    very strong.
3.  Related to the previous point, I do not totally agree that the
    empirical results demonstrate that this approach scales. Again, this
    is because of the assumption about the distance function. In these
    environments, there is a clear and natural distance between agents
    so that this issue is avoided, but in general, deriving such a
    distance function can be possibly *the* main challenge of
    multi-agent RL. By limiting to environments where you can get away
    with this, the demonstration of scalability is limited.

**Additional Comments:**

What assumptions are needed for the mean field Q function to be a
replecament for the standard Q function in Yang et al. (2018)? For
examples, if the agents don't all have the same action space, equation
(3) is not well-defined.

**Audience:**

Yes

**Audience Explanation:**

Multi-agent RL is of course a major area of interest in TMLR, and this
paper presents a method which appears to be quite strong in some
relevant settings.

**Broader Impact Concerns:**

No broader impact concerns.

**Claims And Evidence:**

No

**Claims Explanation:**

See my weaknesses above. Namely, I think the theoretical results paint
the wrong picture of the principles of this algorithm, and I believe
there are some limitations that are largely glossed over which can have
real consequences. However, I think this can be fixed easily in a
revision by being more upfront about the limitations, and by potentially
removing the theoretical results or relegating them to the appendix.

**Requested Changes:**

1.  I want to push back on an assumption made early on. On page 4, it
    says "We assume that such a distance metric is available for the
    domain based on which the importance set is defined". I believe this
    is vastly oversimplifying the problem—perhaps I'm wrong, but this
    definitely merits further discussion. For example, it's easy to
    design a strong deep RL agent when you have a good notion of
    distance between states. But leaning that is the central
    representation learning challenge in RL.
2.  Typo on page 5: "neighbhouhoods".
3.  On page 5, what does "$\overline{a}^j$ approximates agents outside
    the important set" mean? This variable should be defined concretely.
4.  In theorem 1, what does "additively decomposable" mean? Moreover,
    what do you mean by $L$-smooth in the context of action-value
    functions, is this with respect to one of the action inputs? Both of
    the action inputs? State as well?
5.  Theorem 1 itself is pretty weak. It's basically saying you only get
    a good approximation if the Q-function is very flat w.r.t. the
    action.
6.  Similar for Theorem 2. The theoretical results concerning the
    attention basically hold when the Q-function is very insensitive to
    the mean field input.
7.  It should be explicitly clear in the main text how you define the
    distance function to construct the importance sets.

---

> ### Author Response · Authors · 2026-04-22
> **Response to the Reviewer 1/2**
>
> **Theoretical Results:**
>
> We thank the reviewer for this candid feedback and agree with the overall assessment. In response, we have moved the theoretical results to Appendix G, rather than presenting them in the main paper. We agree that the primary contribution of this work is empirical. MFA is a principled and well-motivated framework whose value is demonstrated through extensive experiments across five domains, three testbeds, and hundreds of agents. The theoretical results were originally included to show that prior guarantees for neighbourhood-based MFRL extend to the MFA setting, but we agree that they do not constitute a strong independent theoretical contribution. Relegating them to the appendix better reflects the empirical nature of the paper while keeping them available for readers who wish to verify that the framework is at least consistent with existing theoretical guarantees in the MFRL literature. We have updated the main paper accordingly, removing references to the theoretical results as an independent contribution.
>
> **Assumption on Distance Metric:**
>
> We thank the reviewer for raising this point. We would like to first clarify an important distinction: our paper assumes the availability of a distance metric between agents, not between states. The reviewer's comment references the challenge of learning distance between states, which is indeed a central representation learning challenge in RL. However, agent-to-agent distance is a considerably more accessible quantity — in the vast majority of practical many-agent applications, agents have physical positions or some other natural notion of proximity that is directly observable. In all five of our experimental domains, physical distance between agents is used to construct the importance set, and this information is naturally available as part of the environment's state. This is consistent with several prior works that also rely on locality assumptions between agents, including Qu et al. (2022), Yang et al. (2018), Wang et al. (2022), and Hao et al. (2023), suggesting that this is a broadly accepted and practically reasonable assumption in the MARL community, especially in research concerning many agents environments.
>
> That said, we acknowledge that there may exist domains where even agent-to-agent distance is not straightforward to define, for example, in purely strategic settings where agents do not have physical positions. We have added a discussion of this as a limitation in Appendix A, and note that learning a suitable distance metric or importance weighting from data is an interesting direction for future work.
>
>
>
> **Agents outside importance set:**
>
> We thank the reviewer for this suggestion. We have updated the text to explicitly define the term (please see Eq.5 in the paper). We have defined the mean field $\overline{a}^j$ for MFA as the mean action of all agents outside the importance set of agent $j$, computed as $\overline{a}^j_t = \frac{1}{N - K^j} \sum_{k \notin \mathcal{I}^j} a^k_t$, where $\mathcal{I}^j$ denotes the importance set of agent j, $K^j$ is the size of the importance set ($K^j = |\mathcal{I}^j|$), and $N$ is the total number of agents in the environment. This makes the role of the mean field action as an approximation of agents outside the importance set explicit and concrete.
>
> **Additive Decomposible/L-smooth:**
>
> We thank the reviewer for pointing out these ambiguities. We have added explicit definitions of both "additively decomposable" and "L-smooth" at the beginning of Appendix G, following the same definitions used in Yang et al. (2018). Specifically, additive decomposability refers to the factorization of the multi-agent Q-function into pairwise interactions, and L-smoothness refers to the Lipschitz continuity of the gradient of the Q-function with respect to the actions of other agents. We refer the reviewer to Yang et al. (2018) for a more detailed discussion of these assumptions.
>
> **Theorem 1/Theorem2:**
>
> We agree with the reviewer that both Theorem 1 and Theorem 2 provide relatively weak guarantees. This is precisely why we have moved the theoretical results to the appendix, as they do not constitute a strong independent theoretical contribution. As noted in our response to the first comment regarding the theoretical results, the primary value of these results is not to provide tight bounds but rather to show that the mean field approximation and the convergence guarantees from prior neighbourhood-based MFRL work remain valid even when neighbourhood assumptions are fully relaxed, which is the setting MFA operates in. The strength of MFA as a framework is demonstrated empirically through extensive experiments across five domains and three testbeds, which we believe is the more compelling evidence of its effectiveness.

---

> > ### Author Response · Authors · 2026-04-22
> > **Response to the Reviewer 2/2**
> >
> > **Distance Metric:**
> >
> > We thank the reviewer for this suggestion. We have updated Section 4 to explicitly state that physical distance between agents is used to construct the importance sets in all experimental domains considered in this paper. In our work, for each central agent $j$, the importance set $\mathcal{I}^j$ consists of the $K^j$ agents closest to agent $j$ in terms of Euclidean distance to the central agent. This definition applies consistently across all five experimental domains, including MAgent Battle, Tiger, Combined Arms, Neural MMO, and SMARTS autonomous driving. We also note that while physical distance is the natural and most readily available metric in these domains, MFA is a general framework that can accommodate any domain-appropriate notion of distance or proximity, as discussed in Section 4 and Appendix A.
> >
> > **MFQ assumptions:**
> >
> > The assumptions needed for replacing the multi-agent Q function with the mean field Q function in Yang et al. (2018) are the locality and homogeneity assumptions (described in the last paragraph of Section 2 in the paper). The homogeneity assumption indeed requires that the action space of all agents should be the same. In our work, we have relaxed the requirement of homogeneous agents by using type classification (see Section 4 and Appendix J), as previously done by Subramanian et al. (2020). We demonstrate that MFA methods can be extended to heterogeneous agent environments and still provide better performances as compared to prior methods (as shown in the Combined Arms Experiment in Section 5) using type classification.

---

> > > ### Comment · Reviewer_VDMA · 2026-05-16
> > >
> > > Thanks to the authors for their response and for making the changes proposed by the other reviewers and me, this is much appreciated. I have no further questions.

---

### Review · Reviewer_f3vZ · 2026-04-16

**Summary Of Contributions:**

The paper studies multi-agent reinforcement learning (MARL) problem with large number of agents. As the number of agents increases, the action space gets exponential, which is often handled with mean field approximation argument--the effect of other agents are replaced with the average actions of other agents. The authors propose an attention-based mechanism for local responses and mean-field approximation for global responses.

**Strength**

The authors claim relaxing the homogeneity of the agents and taking consideration of effect of outside neighbourhoods. The authors consider an importance set, and the mean-field approximation is applied to the agents outside this importance set.

The main strength of the paper lies in experimental parts--which verifies the performance of the proposed algorithms with a large number of agents. The experiments are done in MAgent, Neural MMO, and SMARTS environments.


**Weakness**

Theoretical results are provided to justify the proposed framework. Theorem 1 shows that the multi-agent Q-function can be approximated by a mean-field Q-function, and Theorem 2 establishes asymptotic convergence of the proposed algorithm to a Nash equilibrium. However, the technical arguments largely follow standard approximation and convergence analyses in the existing literature, making the theoretical novelty somewhat limited.
Moreover, it is unclear whether the Lipschitz assumption is well aligned with the motivation of the paper. When interactions decay with distance, the influence of distant agents may already be negligible, as discussed in Qu et al. (2022). In such settings, the necessity of applying mean-field approximation outside the neighborhood set is less evident, since the contribution of distant agents may already be sufficiently small without requiring additional approximation structure.



Qu, Guannan, Adam Wierman, and Na Li. "Scalable reinforcement learning for multiagent networked systems." Operations Research 70.6 (2022): 3601-3628.

**Additional Comments:**

na

**Audience:**

Yes

**Audience Explanation:**

Mean-field approximation approach for MARL problem is an important problem as handling exponential action space is difficult. The empirical results provided by the paper would be interesting to the community.

**Claims And Evidence:**

Yes

**Claims Explanation:**

The paper mainly verifies the claims through the experiments. The experiments show that the proposed attention mechanism for multi-agent mean-field algorithm outperforms independent learners and purely mean-field based approaches. Figure 2-4 verifies the results on the MAgent environment. Figure 5 shows results on NMO and SMARTS environment.

**Requested Changes:**

1. The authors could more formally introduce what neighbourood assumptions are.

2. In my opinion, the contributions parts can be more simplified. For example, item 2 can be merged with item 1.

---

> ### Author Response · Authors · 2026-04-22
> **Response to the Reviewer.**
>
> **Theoretical results & influence of distant agents:**
>
> We thank the reviewer for this feedback. In response, we have moved the theoretical results to the appendix (Appendix G) to better reflect the primarily empirical nature of our contributions. We agree that the proof techniques extend those of Yang et al. (2018), and we did not intend to claim novel proof methodology. Rather, our goal was to show that the prior theoretical guarantees for neighbourhood-based MFRL extend naturally to MFA, which relaxes the neighbourhood assumption entirely.
>
> Regarding the reviewer's point about Qu et al. (2022), we have carefully read this work and note that their exponential decay property is explicitly derived under a "local interaction structure," defined by a fixed dependence graph G where the transition distribution of each agent depends only on its local graph neighbourhood. As stated in their paper, this structural assumption is motivated by specific networked system applications such as wireless communication and social networks. This assumption does not hold in the dynamic environments we consider, such as Battle and Tiger, nor in real-world applications like autonomous driving and UAV routing, where agents move freely, interaction structure changes dynamically, and there is no fixed dependence graph governing transitions. In such settings, the influence of distant agents cannot be assumed negligible, and the mean field approximation outside the importance set remains the principled approach for handling them without incurring prohibitive computational cost. We have added a clarifying discussion of this distinction in the paper.
>
> **Neighbourhood Assumptions:**
>
> We thank the reviewer for this suggestion. We have added a formal definition of the neighbourhood assumptions in Section 2 (Background), where we introduce the MFRL framework. This makes the assumptions explicit before we discuss how MFA relaxes them.
>
> **Merging Claim 2 with Claim 1:**
>
>
> We thank the reviewer for this suggestion. We have simplified the contributions section accordingly. Specifically, the description of the two practical MFA algorithms (MFA-QL and MFA-AC) has been merged with the description of the MFA framework itself into a single contribution item, as suggested.

---

### Author Response · Authors · 2026-04-22
**Thanks to the reviewers and summary of changes.**

We thank all the reviewers for their detailed review of the paper. We have submitted a revision of the paper with the following changes:


- Based on the comments from the reviewers, we have moved the two theorems in the paper to the appendix and modified the claims accordingly. All the existing claims in the paper are supported by the experiments. This addresses the comments of Reviewer VDMA and Reviewer f3vZ.

- We have introduced one additional appendix (Appendix K) containing experiments on action advising as applied to three best performing baseline algorithms from prior works in the MAgent Battle domain. This is to test the performance of these baselines incorporating action advising as done for the MFA algorithms. This addresses the comments provided by Reviewer  DuYF.


- We have included two additional baseline algorithms (MFQ and MFAC) and one of our methods (MFA-QL) for performance comparisons in the Experiment 3 (which uses a heterogeneous agent environment). This addresses the comments provided by Reviewer DuYF.


- All the typos and minor comments mentioned by the reviewers have been fixed in the paper.

Furthermore, we provide our detailed responses to each question and comment of the reviewers below. We believe that these changes have addressed all the comments from the reviewers. If there are any additional questions or comments, we welcome the opportunity to address those comments as well.

---

### Decision · Action_Editor_X1RY · 2026-06-04

**Recommendation:** Accept with minor revision

**Additional Comments:**

The paper meets the TMLR bar after revision, but the final version may benefit from making  the remaining limitations and empirical framing especially clear to ensure the following before publication
- The main text clearly states that the primary contribution is empirical and methodological, rather than a strong new theoretical contribution.
- The theoretical results are presented as supporting consistency with prior mean-field guarantees, not as a central independent contribution.
- The construction of the importance set, including the role of the distance metric and the choice of set size K, is clearly described in the main paper, along with the limitation that these choices may be domain-dependent.
- The discussion of reward versus win-rate evaluation is made easy to follow, especially where execution performance is evaluated against a fixed strong opponent.
- The added heterogeneous-agent baselines, action-advising comparisons, and ablations are clearly referenced from the main text or appendix so that readers can see how the revision addresses the reviewers’ concerns.
- The authors include a concise limitations paragraph noting that MFA is most compelling in settings where local interaction structure is meaningful and where a reasonable distance or proximity notion is available.

**Audience:**

Yes

**Audience Explanation:**

Yes. The paper addresses an important scalability problem in multi-agent reinforcement learning, which is clearly relevant to a portion of the TMLR audience. Mean-field methods are a common approach for scaling MARL, and this paper studies a practically useful extension that retains scalability while incorporating richer local interaction information through attention. The empirical results, ablations, and discussion of heterogeneous-agent settings should be of interest to researchers working on MARL, mean-field RL, large-population agent systems, and practical algorithms for environments with many interacting agents.

**Claims And Evidence:**

Yes

**Claims Explanation:**

Yes. The paper’s main claims are now adequately supported by the empirical evidence and by the revisions made during the review process. The proposed MFA framework is positioned as a practical relaxation of standard mean-field MARL assumptions, combining attention over an importance set of local agents with a mean-field approximation for the remaining agents. The reviewers generally agreed that the experimental results show consistent empirical benefits over relevant baselines across several multi-agent environments, and the added ablations and comparisons in the revision help clarify the source of the gains.

The main concerns raised during review concerned the strength of the theoretical claims, the role of the distance metric/importance set, heterogeneous-agent comparisons, action advising, and the interpretation of reward versus win-rate results. The authors addressed these issues by weakening the theoretical claims, moving the theory to the appendix, clarifying the neighborhood and distance assumptions, adding additional baselines and ablations, and expanding the discussion of limitations. While the theoretical contribution remains modest and some design choices are heuristic or domain-dependent, the claims in the revised paper are appropriately framed as primarily empirical and are supported to the standard expected for TMLR.